# Pressurised water flow in fractured permafrost rocks revealed by borehole temperature, electrical resistivity tomography, and piezometric pressure

Maike Offer[1], Samuel Weber[2,3], Michael Krautblatter[1], Ingo Hartmeyer[4], and Markus Keuschnig[4]

[1]Chair of Landslide Research, Technical University of Munich, Munich, Germany
[2]WSL Institute for Snow and Avalanche Research SLF, Davos, Switzerland
[3]Climate Change, Extremes and Natural Hazards in Alpine Regions Research Center CERC
[4]GEORESEARCH, Puch bei Hallein, Austria

**Correspondence:** Maike Offer (maike.offer@tum.de)

**Abstract.** Rock slope instabilities and failures from permafrost rocks are among the most significant alpine hazards in a changing climate and represent considerable threats to high-alpine infrastructure. While permafrost degradation is commonly attributed to rising air temperature and slow thermal heat propagation in rocks, the profound impact of water flow in bedrock permafrost on warming processes is increasingly recognized. However, quantifying the role of water flow remains challenging, primarily due to the complexities associated with direct observation and the transient nature of water dynamics in rock slope systems. To overcome the lack of quantitative assessment that inhibits thermal and mechanical modelling, we perform a joint analysis of borehole temperature, electrical resistivity measurements, and piezometric pressure. Therefore, we combine datasets of rock temperature measured in two deep boreholes (2016-2023), monthly repeated electrical resistivity tomography acquired in 2013 and 2023, site-specific temperature-resistivity relation determined in laboratory with samples from the study area, and borehole piezometer. Field measurements were carried out at the permafrost-affected north flank of the Kitzsteinhorn (Hohe Tauern range, Austria), characterized by significant water outflow from open fractures during the melt season. Borehole temperature data demonstrate a seasonal maximum of the permafrost active layer of 4-5 m. They further show abrupt temperature changes ($\sim$ 0.2-0.7 °C) during periods with enhanced water flow and temperature regime changes between 2016-2019 and 2020-2022, which cannot be explained solely by conductive heat transfer. Monthly repeated electrical resistivity measurements reveal a massive decrease in resistivity from June to July and the initiation of a low-resistivity ($< 4$ k$\Omega$m) zone in the lower part of the rock slope in June, gradually expanding to higher rock slope sections until September. We hypothesize that the reduction in electrical resistivity of more than one order of magnitude, which coincides with abrupt changes in borehole temperature and periods of high water heads up to 11.8 m, provides certain evidence of snowmelt water infiltration into the rockwall becoming pressurised within a widespread fracture network during the thawing season. This study provides for the first time direct and indirect field measurements of pressurised water flow which shows that, in addition to slow thermal heat conduction, permafrost rocks are subjected to sudden push-like warming events and long-lasting rock temperature regime changes, favoring accelerated bottom-up permafrost degradation, and contributing to the build-up of hydrostatic pressures potentially critical for rock slope stability.

# 1 Introduction

Over the past decade, awareness of climate change-related hazards in alpine environments has increased, including rock slope failures from warming permafrost rocks. Degrading permafrost in rock walls can reduce the stability of rock slopes (Gruber and Haeberli, 2007), increasing the likelihood of slope failure and endangering mountain infrastructure (Duvillard et al., 2021b). Rising temperatures alter the mechanical properties of rock and ice (Krautblatter et al., 2013; Mamot et al., 2018), which can increase the number of rockfalls and rock-slope failures in permafrost-affected rockwalls (Ravanel et al., 2017; Hartmeyer et al., 2020; Savi et al., 2021; Cathala et al., 2024b). Therefore, precise knowledge of changes in frozen rock's thermal and mechanical regime is of particular interest to address mountain tourism's increased vulnerability and prevent or limit damage to high-alpine infrastructure (Duvillard et al., 2019).

Potential triggers of rock slope failure have been attributed to high water pressures (e.g., Fischer et al., 2010; Ravanel et al., 2017) that can arise from the infiltration of rainwater or meltwater from snow, glaciers, ice-filled joints or permafrost. In some cases, the hydrological processes contributing to the destabilization of the rock slope were directly evidenced by ice and water coating the scarp shortly after the event (e.g., Walter et al., 2020; Mergili et al., 2020; Cathala et al., 2024a). The infiltration capacity and hydraulic permeability of a rockwall are determined by the degree of fracturing, pore space characteristics, saturation, and temperature (Gruber and Haeberli, 2007). The surface water availability from snowmelt and rainfall for infiltration into steep rock slopes was recently estimated by a numerical energy and hydrological balance model (Ben-Asher et al., 2023). Ice-filled fissures reduce the permeability compared to unfrozen fissures (Pogrebiskiy and Chernyshev, 1977), leading to perched water and the build-up of high hydrostatic pressures (Magnin and Josnin, 2021). Although the destabilizing shear stress of high water levels on the rock mass is well known (Krautblatter et al., 2013), piezometric pressures have only been observed on one rock glacier (Phillips et al., 2023; Bast et al., 2024) and in one open crack at shallow depth (Draebing et al., 2017). Direct observations of piezometric pressures in deep depths (> 10 m) have not yet been measured in permafrost-affected rockwalls, but remain crucial for understanding hydrological processes and thus for the prospective prediction of rock slope failures.

Non-conductive heat transport due to water flow along fractures can rapidly raise the temperature, leading to the formation of thaw corridors, which potentially enhance local permafrost degradation (Gruber and Haeberli, 2007) and erosion of ice-filled clefts (Hasler et al., 2011a) compared to slow thermal heat propagation in rocks. The physical processes of water flux in frozen bedrock are intrinsically difficult to quantify, exhibit non-linear, complex behavior, and are therefore poorly studied. Phillips et al. (2016) observed brief and intermittent thermal anomalies in borehole measurements in an alpine rock ridge, interpreted as water percolating through fractures after heavy precipitation. Seismic analysis and geophysical soundings have revealed seasonal variations in cleft-ice infill and meltwater percolation in fractures (Krautblatter et al., 2010; Keuschnig et al., 2017; Weber et al., 2018).

Changes in water and ice content can be detected by electrical resistivity tomography (ERT), which is, therefore a well-established method (e.g., Hauck, 2002; Schneider et al., 2013; Hilbich et al., 2022; Cimpoiasu et al., 2024) that has been increasingly applied in mountain permafrost environments in recent years (Herring et al., 2023). Monitoring active layer dy-

namics and spatial and temporal permafrost evolution requires repeated ERT surveys, preferably using the same survey geometry and fixed installed electrodes. ERT monitoring over several years has demonstrated degrading permafrost associated with global warming (e.g., Mollaret et al., 2019; Scandroglio et al., 2021; Buckel et al., 2023). However, most monitoring approaches emphasize the inter-annual evolution of bedrock permafrost and often overlook seasonal variations in resistivity, especially during thawing periods.

ERT measurements offer valuable, indirect information, requiring either laboratory calibration (Krautblatter et al., 2010) or ground temperature for quantitative interpretation. Borehole temperature measurements are direct thermal observations. They only provide point-scale data along the borehole profile in a heterogeneous alpine terrain but do not allow for differentiation between water and ice close to 0 °C. The freezing point of water can be shifted depending on rock properties, salinity, and pore pressure (Arenson et al., 2022), allowing unfrozen water content at sub-zero temperatures. In addition, temperature changes may be slowed by latent heat effects while significant changes in the water-to-ice ratio occur. To reduce uncertainties in the characterization of subsurface properties, a complementary approach that combines borehole observation and geophysical data such as electrical resistivity (e.g., Hauck, 2002; Etzelmüller et al., 2020) and/or seismic refraction data (e.g., Hilbich, 2010; Wagner et al., 2019) is required.

The influence of water fluxes on the thermal and hydrostatic regime in bedrock permafrost remains a key challenge, despite the demonstrated link between hydrological fluxes and rock slope instability. In this study, we address the lack of quantitative and in-situ observations of rockwall hydrology by analyzing repeated ERT measurements, ground temperature data from two deep boreholes (2016-2023) and piezometric pressure data (2024) at the fractured north face of the Kitzsteinhorn (Hohe Tauern range, Austria). This study highlights the importance of a higher geoelectrical measurement frequency compared to annual campaigns, an integrated approach of different indirect and direct measurements, and emphasizing the significance of melt period analysis in delineating both temporal patterns and spatial distribution of intense water flow within fractures. These insights will improve our knowledge of the complex seasonal water flow in rockwalls that potentially accelerate permafrost degradation and contribute to promoting and triggering factors for rock slope failures.

## 2  Study site

The investigated rockwall is located in the summit region of the Kitzsteinhorn (3.203 m asl), part of the Hohe Tauern range in Austria (Fig. 1). The Kitzsteinhorn region is characterized by rocks of the Penninic Bündner Schist Formation in the Tauern Window and belongs to the Glockner Nappe. The lithology in the study area is dominated by calcareous mica-schists traversed by a scaly band of serpentinite parallel to the schistosity that dips steeply with approximately 45° to NNE (Cornelius and Clar, 1935; Hoeck et al., 1994; Hartmeyer et al., 2020).

The north face of the Kitzsteinhorn exhibits a significant degree of fracturing, characterized by joint openings of up to 20 cm, predominantly along cleavage planes (Schober et al., 2012). The development of the enormous number of joint sets was favored by stress release and intense physical weathering processes (Hartmeyer et al., 2012). The main joint sets are K1, which has a sub-vertical dip to the west, and K2, which features a steep dip to the southwest (Fig. 1). K3 and K4 are less

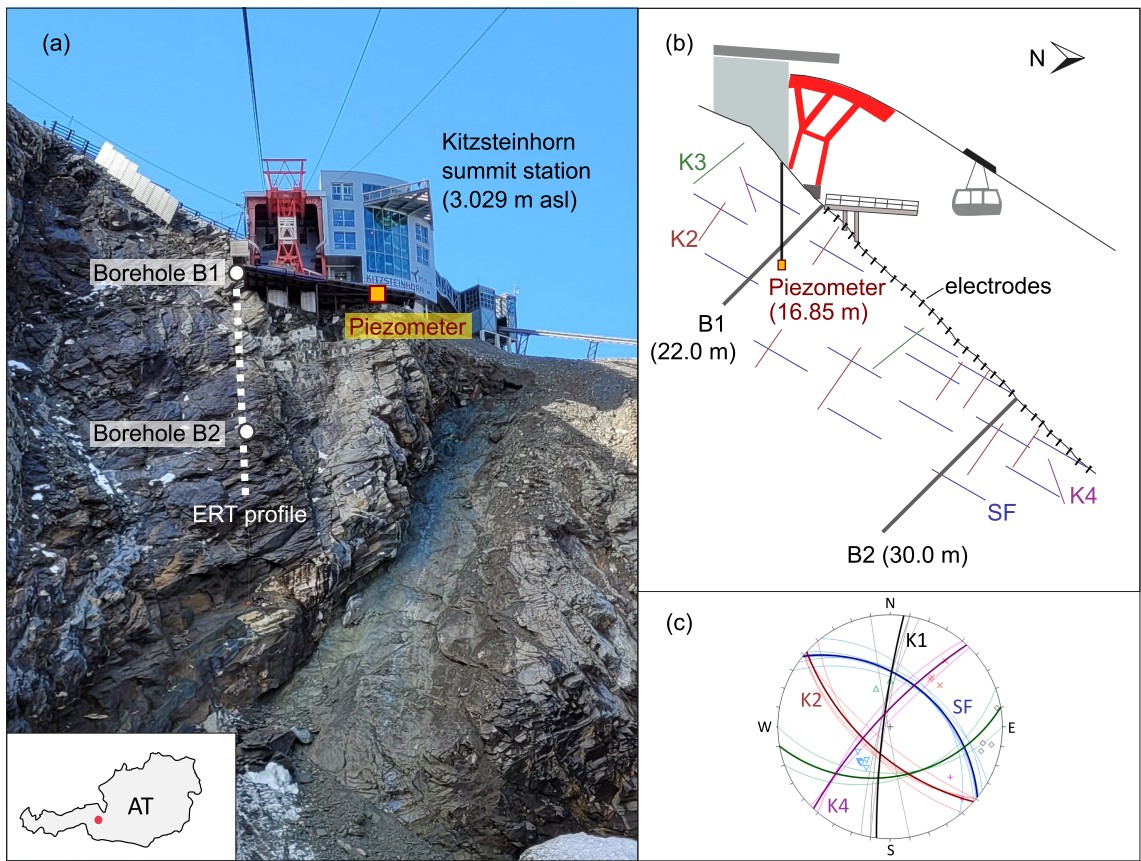

**Figure 1.** (a) Detailed view of the investigated, north faced rockwall below the Kitzsteinhorn summit station (Hohe Tauern range, Austria), the position of the ERT profile, the two deep boreholes (B1/B2), and the piezometer. (b) The geotechnical setting of the rock face is described by the schematic representation of the main discontinuities K1, K2, K3, K4, and the schistosity SF, as well as (c) the dip angle and direction of their mean set planes.

abundant. K3 dips medium-steeply to flat to S-SSE, and K4 dips steeply to NW. Laboratory tests of mica-schist rocks with ice-filled joints have shown that the shear resistance along rock-ice interfaces follows a temperature- and stress-dependent failure criterion, where warming and unloading lead to a mechanical destabilization and can lead to self-enforced rock slope failure (Mamot et al., 2020). Keuschnig et al. (2017) installed 2012 the ERT profile at the north-facing rockwall below the station of the Kitzsteinhorn cable car, ranging from an elevation of 2.965 to 3.017 m asl (Fig. 1). The flank has been severely affected at its base by the rapid retreat of the Schmiedingerkees Glacier, which caused increased rockfall activity in the lower part of the rockwall, with single detachments of up to 880 m³ (Hartmeyer et al., 2020). Several rockfall events have damaged the lower part of the ERT profile. As a result, the electrical resistivity measurements, carried out monthly from June to September 2023, now cover an elevation range from 2.976 to 3.017 m asl. The slope gradient of the rock face increases as it approaches the glacial cirque and is averaged at 47° (Keuschnig et al., 2017). Weather conditions at the study site in 2013 and 2023 were

analyzed using recordings from the Glacier Plateau and Alpincenter weather stations, including air temperature, snow height, and rainfall (Fig. A1).

This study is part of the 'Open-Air-Lab Kitzsteinhorn' (Keuschnig et al., 2011; Hartmeyer et al., 2012), a comprehensive long-term project to monitor the response of bedrock permafrost to climatic change across multiple scales. Empirical-statistical calculations by Schrott et al. (2012) on the distribution of permafrost in the Hohe Tauern region indicate a high permafrost probability above 2.500 m asl on north-facing rockwalls and above 3.000 m asl on south-facing slopes of the Kitzsteinhorn. In addition, permafrost has been detected by electrical resistivity measurements in the northern section of the tunnel 'Hannah-Stollen', which connects the Kitzsteinhorn cable car station to the south side at an elevation of approximately 3.000 m asl (Hartmeyer et al., 2012).

## 3 Methods and data processing

### 3.1 Borehole temperature measurements

Rock temperature measurements in two deep boreholes close to the geoelectrical profile have been conducted since December 2015 and were analyzed until December 2023. Borehole B1 is situated close to electrode 1 (x = 0 m) at an elevation of 3.030 m asl, and borehole B2 is positioned between electrodes 22 (x = 42 m) and 23 (x = 44 m) at an elevation of 2.970 m asl. Both boreholes were drilled perpendicular to the surface and the schistosity, reaching a depth of 22 m (B1) and 30 m (B2). The temperature inside both boreholes was recorded based on a highly sensitive measuring system: After drilling, B1 and B2 were equipped with water- and airtight polyethylene casings interrupted by non-corrosive brass rings at the designated depths of the temperature sensors. The annulus was filled with concrete to avoid water/air circulation and ice formation in the void between the casing and ambient bedrock. A chain of customized high-precision Pt100 thermistors (accuracy $\pm$ 0.03 °C) was inserted into the casing and established direct mechanical contact with the brass rings of the casings. Due to the high thermal conductivity of the brass rings and their direct physical connection to the sensors, this setup enables improved thermal coupling between the temperature sensor and surrounding rock mass compared to most other systems. The system used for the present study thus responds sensitively to (minor) changes in temperature and is well-suited to record short-term fluctuations of bedrock temperature. Sensors were installed at 0.1, 0.5, 1, 2, 3, 5, 7, 10, 15, 20, 21.5 (only B1), 25 (only B2) and 30 m (only B2) depth. The measuring interval was set to 10 minutes. Scaled images of the borehole walls were acquired using an optical scanner to identify and locate discontinuities. These images provided insight into parameters including joint frequency and aperture over the entire depth of B1 (22 m) and to a depth of approximately 15 m in B2.

Lightning strikes damaged several thermistors throughout the long-term monitoring (2016-2023), leading to data gaps starting in June 2017 (B2 = 0.5, 10 m), July 2019 (B2 = 20 m), June 2020 (B2 = 7 m), September 2020 (B2 = 25 m), April 2023 (B1 = 20 m), June 2023 (B1 = 7, 10, 21.5 m, B2 = 1 m). Warming releases resulting from construction activities for summit station maintenance in summer 2023, mainly due to drilling operations near B1, could have affected ground temperature measurements. Consequently, we excluded the affected data sets from B1 (August-December 2023).

The thermistor signals in B1 and B2 were analyzed for irregularities and characteristics typical for non-conductive heat transfer. Near-surface temperatures (depth < 2 m) were excluded from the analysis as they are characterized by short-term fluctuations with large amplitudes, making distinguishing between changes induced by non-conductive heat transfer and meteorologically forced changes complex. For the recordings in 2 and 3 m depth, thermal anomalies were identified using the first derivative with a moving average of 12 points (i.e., measurement interval of 2 hours). High signals in the first derivative were manually reviewed for characteristics typical of non-conductive heat transfer, which exhibit a temperature rise of up to 0.7 °C in less than 2 hours. Sudden, significant changes between two measurements (10 min) with a return to the previous temperature level are caused by overvoltage effects following lightning strikes and were therefore not considered further. Due to the smooth curvature of the thermal signals in 5 m depth, thermal anomalies were directly visible in the data and were manually determined.

## 3.2 ERT data acquisition

ERT is a non-invasive geophysical method for detecting and monitoring the distribution and properties of permafrost: an electric current is injected into the subsurface by two electrodes, and the resultant electrical potential difference is measured at two other electrodes. The respective apparent resistivity of the electrical field is then determined by the injected current, the measured potential difference, and a geometric factor. The investigation depth depends on the electrode spacing, with shorter spacing resulting in a lower depth of investigation but higher resolution near the surface. The apparent resistivity values need to be inverted using an appropriate algorithm to provide areal information on the electrical resistivity distribution. Ice-filled joints and frozen or dry bedrock are typically characterized by high electrical resistivity, whereas unfrozen conditions and high liquid water content are associated with low electrical resistivity (Hauck, 2002). The ERT profile used for the present study was installed in 2012 and initially used for a year-long continuous subsurface monitoring of permafrost conditions using automated ERT starting in February 2013 (Keuschnig et al., 2017). Therefore, 37 electrodes were permanently drilled into the bedrock at intervals of 2 m. Stainless-steel climbing bolts with a length of 90 mm and a diameter of 10 mm were preferred to fixed stainless-steel screws (Krautblatter and Hauck, 2007) as they allowed good electrical contact with the bedrock without applying metallic grease and ensured long-term durability. The automated ERT measurements were performed from February-December 2013 on a 4-hour sampling interval using Wenner configuration, as it provides the best signal-to-noise ratio in permafrost rockwalls (Zhou and Dahlin, 2003) and good horizontal resolution (Hauck and Kneisel, 2008). Two lightning strikes severely damaged the hardware and prevented most measurements between June and September 2013. We performed ERT measurements over a 24-hour period at the end of each month from June to September 2023 using the same protocol type to fill this data gap. We analyzed short-term variations in electrical resistivity values for each measurement campaign to exclude random errors and quantify the respective measurement uncertainty (Fig. B1). Due to several rockfall events, the lower part of the ERT profile was damaged, and only the top 30 fixed electrodes could be used in 2023 (i.e., 58 m profile length). To enable comparison of the 2013 and 2023 data series on a temporal and spatial scale, we selected tomographies at the end of each month of the automated ERT measurements in 2013 (February-June and September-December) and shortened them to the same profile length (electrodes 1-30). Further technical details of the respective ERT measurements are given in Table B1.

The ERT measurements conducted in 2023 were carried out under varying weather conditions (Fig. 2). A thin layer of ice

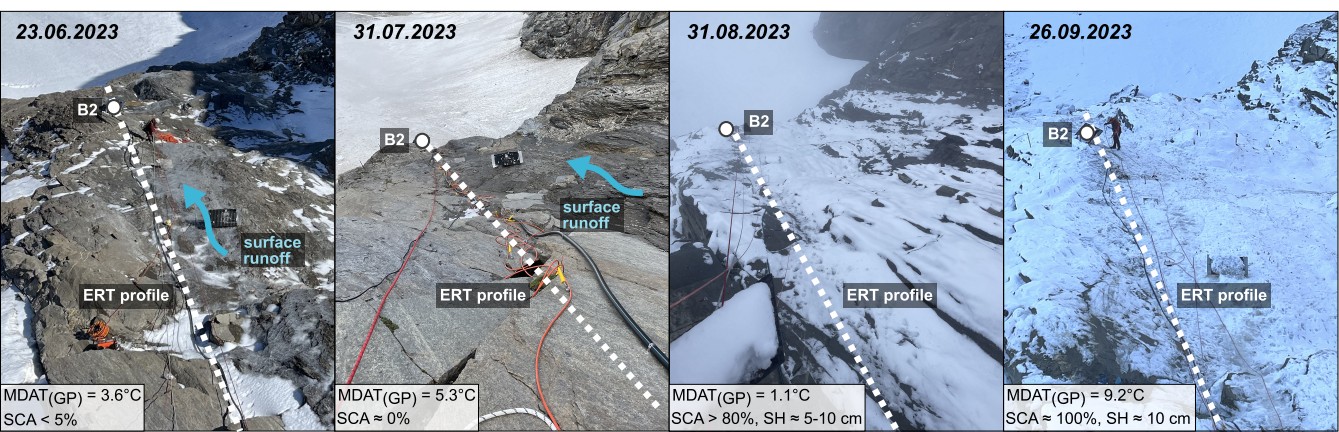

**Figure 2.** Weather and rock surface conditions on the days of ERT measurements in 2023 described by mean daily air temperature (MDAT) derived from the nearby ($\sim$ 500 m distance) weather station at the Glacier Plateau (GP), and estimated values of snow cover area (SCA) and snow height (SH). During the ERT measurements in June and July, pronounced surface runoff was observed in the lower part of the ERT profile.

partially covered the rock surface during the morning and night of the ERT measurements in June, while significant surface runoff was observed passing the lower part of the ERT profile during the afternoon of the June campaign and throughout the measurement period in July. During ERT observations in August and September, most of the rockwall was covered by snow, reaching a height of up to 10 cm. Mean daily air temperature derived from a nearby weather station on the Glacier Plateau (distance $\sim$ 500 m, 2.940 m asl) indicated positive mean values for all days of ERT measurements in 2023. Datum points with considerable deviations from neighboring values were manually filtered, resulting in a maximum declaration of 2 % of datum points per measurement. These outliers were masked and excluded from further data processing. All ERT measurements were inverted using the software package Res2Dinv (Loke and Barker, 1996) with adapted settings for permafrost rock faces. Robust inversion routines with half electrode spacing and mesh refinement were applied to cope with expected high electrical resistivity gradients (Krautblatter and Hauck, 2007). After four to five iterations, a reasonable level of convergence was achieved without over-fitting the data. Resistivity models with a high root mean square (RMS) error between the modelled and observed data are obtained, particularly during the winter months when frozen surface conditions impede the coupling of electrodes. Contact resistance values vary seasonally (Herring et al., 2023); in our case, values < 190 k$\Omega$ were observed during summer measurements, while values > 200 k$\Omega$ were mainly measured during winter measurements (Table B1). The high contact resistances (> 200 k$\Omega$) could not be mitigated due to safety issues of accessing the problematic electrodes during the snow-covered period of the rock face. However, an assumption of inaccuracy and subsequent complete exclusion of the affected data sets, justified by high contact resistances and high RMS errors, would make it impossible to provide all-season ERT observations. We, therefore, retained the data sets of the winter measurements (February-May and September-December 2013) but withheld

detailed interpretation in recognition of the potential presence of noise. In addition, the change in the raw data, i.e. the apparent resistivity, of the different measurements (June 2013, June-September 2023) was analyzed in relation to the investigation depths to exclude artifacts from the inversion (Fig. B2). A summary of the data post-processing is given in the Appendix (Table B1), as well as the differences in the resistivity between the tomograms from June-September 2023 (Fig. B3).

### 3.3 Laboratory calibration of temperature-resistivity relation

The temperature-resistivity relation was derived from laboratory experiments of seven cylindrical samples ($\varnothing = 80$ mm) drilled from four calcareous mica-schist rock samples from the study side. The core samples were saturated for several weeks with tap water. At the chemical equilibrium ($\sim 17$ °C), water conductivity was $0.058 \pm 0.002$ S m$^{-1}$, which is in the same order of magnitude as snowmelt water from the study site, with a conductivity of $0.014$ S m$^{-1}$. Full saturation of pore spaces was achieved under atmospheric pressure, as applying a vacuum would result in higher water absorption, which is less representative of field conditions (Krus, 1995; Sass, 1998). However, the free saturation method is likely subject to the occurrence of air bubbles (Draebing and Krautblatter, 2012), which may complicate the interpretation of the results.

Figure 3a illustrates the experimental setup for the laboratory electrical resistivity measurements. Two copper plates were

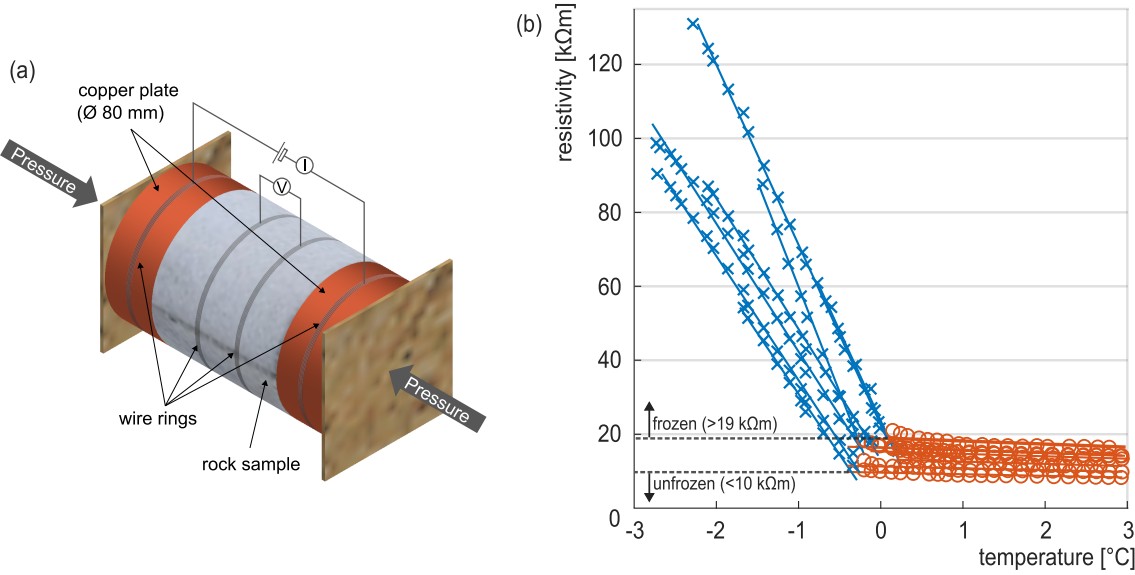

**Figure 3.** (a) Simplified schematic illustrating the setup of the laboratory calibration. (b) Temperature-resistivity relation of laboratory experiments of seven mica-schist core samples indicating unfrozen conditions under 10 kΩm and frozen conditions above 19 kΩm.

placed on the grounded end-faces of the samples as current electrodes (A and B). For the voltage electrodes (M and N), copper pipe clamps were installed around the core sample with equal spacing between the electrodes. Pressure was applied to the current electrodes to ensure good electrical contact. Resistivity $\rho$ was calculated from the cross-section of the core sample $A$,

the applied voltage $V$, the distance between the voltage electrodes $L$, and the resulting current $I$ (Telford et al., 2004):

$$\rho = \frac{AV}{LI} \tag{1}$$

Temperature of the core samples was controlled by a freezing chamber which can be regulated in 0.1 °C increments. The rock samples were slowly warmed from $-3$ to $3$ °C with a gradient of $0.3$ °C h$^{-1}$ to ensure temperature homogeneity. To avoid electrical interference, rock temperature was measured on a second, characteristically identical sample at depths of 25 and 40 mm with two thermometers (Greisinger GMH 3750, with 0.1 °C accuracy GTF 401 sensors). The samples were loosely covered with plastic film and ventilated to prevent drying and thermal layering. According to the field investigations, continuous resistance measurements were done with ABEM terrameter LS using a minimum current of 0.1 mA and a maximum voltage output of 600 V.

The electric properties of water-saturated rocks is determined by the ionic transport in the liquid phase and, therefore, by the amount of interconnected pores. The well-known empirical law develop by Archie (Archie, 1942) relates the resistivity $\rho$ to the functional porosity $\phi$, the resistivity of the pore water $\rho_w$, and the fraction of the pore space occupied by liquid water $S$:

$$\rho = a\phi^{-m}S^{-n}\rho_w \tag{2}$$

where $a$, $n$ and $m$ are empirically determined constants. At subzero temperatures and under partially frozen conditions, the electrical properties of the rock depends on the remaining unfrozen water content in the pores. As the temperature drops to the equilibrium freezing temperature, pore water saturation decreases while the resistivity of the pore water also decreases due to the migration of electrolytes from the freezing water to the remaining unfrozen water content, resulting in increased electrolyte concentration. Above the equilibrium freezing temperature, the resistivity of the rock is indirectly related to temperature changes, as temperature affects the mobility of the solute electrolytes.

### 3.4 Piezometer installation

The piezometer sensor GEOKON 4500S (accuracy $\pm 0.1$ % F.S. from 0-7 bar) was installed in a vertical borehole ($\varnothing = 90$ mm, 20.5 m depth) close to the summit station (Fig. 1). The desired piezometer tip location was determined by the scaly band of serpentinite, identified through optical scanning of the borehole walls between 16.85 to 17.70 m depth. Consequently, the bottom of the borehole was filled with impermeable cement, followed by the isolated collection zone consisting of the piezometer sensor positioned within a bag of filter sand ($\varnothing = 90$ mm, $\sim 30$ cm length) precisely at the upper boundary surface of the calcareous mica-schists and serpentinite at a depth of 16.85 m. The cavity between the porous filter stone and the diaphragm was filled with a water-glycol mixture to prevent freezing damage to the piezometer diaphragm. The borehole above the water permeable collection zone was subsequently sealed with impermeable bentonite and cement.

A thermistor integrated into the housing of the piezometer sensor measured the temperature (accuracy $\pm 0.5$ °C between -80 and +150 °C). The vibrating wire and thermistor are connected to a wireless Worldsensing data logger LS-G6-VW, which includes a barometer (relative accuracy $\pm 12$ hPa), and transmits the hourly measured data through LoRa communications. The

pressure data was calculated with temperature and barometric corrections using the equation:

$$P_{corr} = (R - R_0)G + (T - T_0)K - (S - S_0) \tag{3}$$

where $R$ is the measured pressure reading, $R_0$ is the initial zero pressure reading, $G$ is the linear calibration factor, $T$ is the measured temperature, $T_0$ is the initial zero temperature, $K$ is the thermal factor, $S$ is the measured barometric pressure and $S_0$ is the initial zero barometric pressure. The recording began at the end of October 2023; however, to exclude long-lasting warming effects from the drilling activities, we analyzed the data set starting from January 2024 until mid of June 2024.

## 4 Results

### 4.1 Borehole temperature and thermal anomalies

Interpolated borehole temperature records from ERT measurement days in 2023 indicate subzero subsurface conditions below depths of 0.5 m (B1) and 1.7 m (B2) in June, 3.3 m (B1) and 2.7 m (B2) in July, 3.4 m in August (B2), and 3.5 m in September (B2), see Fig. 4a. The annual variations in ground temperature as well as the characteristic parameters of active

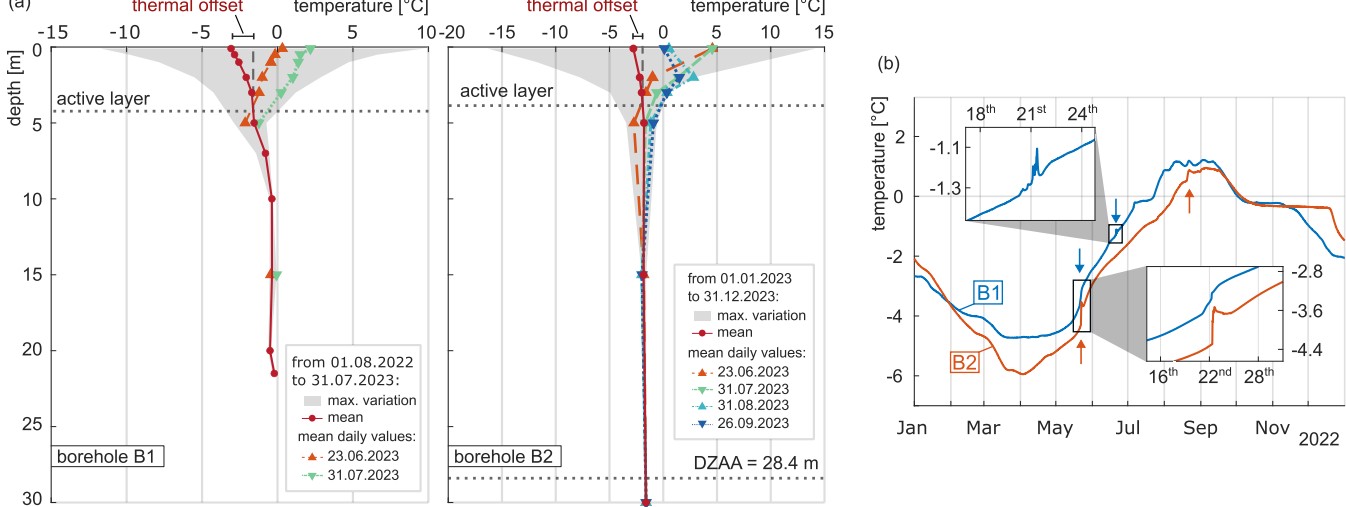

**Figure 4.** (a) Ground temperature in boreholes B1 and B2 at the ERT measurement days from June to September 2023. The maximum annual variation of the ground temperature, the thickness of the active layer, the depth of zero annual amplitude (DZZA), and the mean annual ground temperature were determined for representative periods, respectively (B1: the year before construction activities for summit station maintenance close to the borehole started in August 2023, B2: the year 2023). (b) Thermal anomalies (abrupt temperature changes) found in the borehole temperature data, exemplified by the record taken at 3 m depth from 2022.

layer thickness (ALT), depth of zero annual amplitude (DZAA), and thermal offset were calculated for B2 for 2023, while for
B1, the year before construction activities for summit station maintenance close to the borehole (August 2022-July 2023) was

analyzed. The seasonal maximum of the ALT was reached at a depth of 4.3 m in B1 and of 3.9 m in B2; the DZAA, defined by the annual temperature amplitude of less than 0.1 °C (Gruber et al., 2004), was only reached in B2 at around 28.4 m. The thermal offset, defined by the difference between the annual mean ground surface temperature (i.e., temperature recording in 0.1 m depth) and the temperature at the permafrost table (Burn and Smith, 1988), is generally considered impracticable for fractured bedrock due to the high variable microclimate and active layer thickness (Hasler et al., 2011b). However, in our specific case, both boreholes suggest that the permafrost table depth are within a similar range. Therefore, considering the potential variability in microclimate, we propose that the concept of thermal offset is practicable and demonstrate positive values (B1 = 1.5 °C, B2 = 0.9 °C).

The quasi-sinusoidal temperature curves recorded over the observation period at 2, 3, 5, 10, and 15 m depth suggest that heat transport in the intact rock mass is dominated by conduction (Fig. 5). Nonetheless, borehole temperature showed 69 abrupt changes (thermal anomalies) at 2 and 3 m depth (B1, B2). Borehole temperatures at greater depths from 5-30 m are likely less sensitive to heat transfer from percolating water, but exhibited 13 irregularities at 5 m depth (B1, B2) and notable changes in the quasi-sinusoidal pattern since 2020 at 10 and 15 m depth (B1). All abrupt changes and irregularities in the borehole temperature occurred between late May and September, coinciding with the significant decrease in resistivity observed in the ERT measurements from June to September (Fig. 7).

Optical borehole imaging shows a pronounced occurrence of clefts with apertures of up to 5 cm in the first ten meters in B1 and B2 and intact rock mass of calcareous mica-schist with marked schistosity at greater depths (Fig. 5). Thermistors installed close or within clefts or areas of schistosity exhibit a higher frequency of thermal anomalies, as evidenced by the counts recorded (e.g., B1-2 m: 18, B2-3 m: 25), in contrast to thermistors installed at a greater distance, not exceeding 50 cm, from discontinuities (B2-2 m: 10, B1-3 m: 16). Most of the thermal anomalies result in the warming of the rock mass, while only two events in August and September 2017 induced cooling in B2 at depths of 3 m. The initial rock temperature shortly before rapid changes at 2 and 3 m depth (B1 and B2) indicates that the push-like warming and cooling events were recorded mostly during frozen conditions of the surrounding rock mass (72 % of thermal anomalies, Fig. C1).

Long-term changes in the temperature regime were observed between 2016-2019 and 2020-2022 at depths of 10 and 15 m in borehole B1. These changes were characterized by a quasi-sinusoidal pattern of rock temperature from 2016-2019, followed by rapid and irregular warming between 2020-2022 (Fig. 6a). Minimal annual rock temperatures at 10 m depth ranged from -1.34 to -0.94 °C from 2016-2019, increasing to -0.73 to -0.49 °C from 2020-2022. Maximal annual rock temperatures were observed in winter/spring from 2016-2019, ranging from -0.85 to -0.70 °C, and shifted to summer months from 2020-2022, with temperatures between -0.23 and -0.06 °C. The most pronounced changes in mean monthly rock temperatures were recorded in July in 10 m depth (2017: -1.31 °C, 2020: -0.16 °C) and in September in 15 m depth (2017: -0.95 °C, 2022: -0.10 °C), see Fig. 6b.

The polar plot in the center of Fig. 7 shows a one-year bedrock temperature cycle measured in B2 from January to December 2023. Rock temperature at depth shows a delayed and damped response to cooling and warming. At B2, for instance, a temperature maximum at 5 m was recorded in late December (−0.6 °C) while the minimum was registered in early May

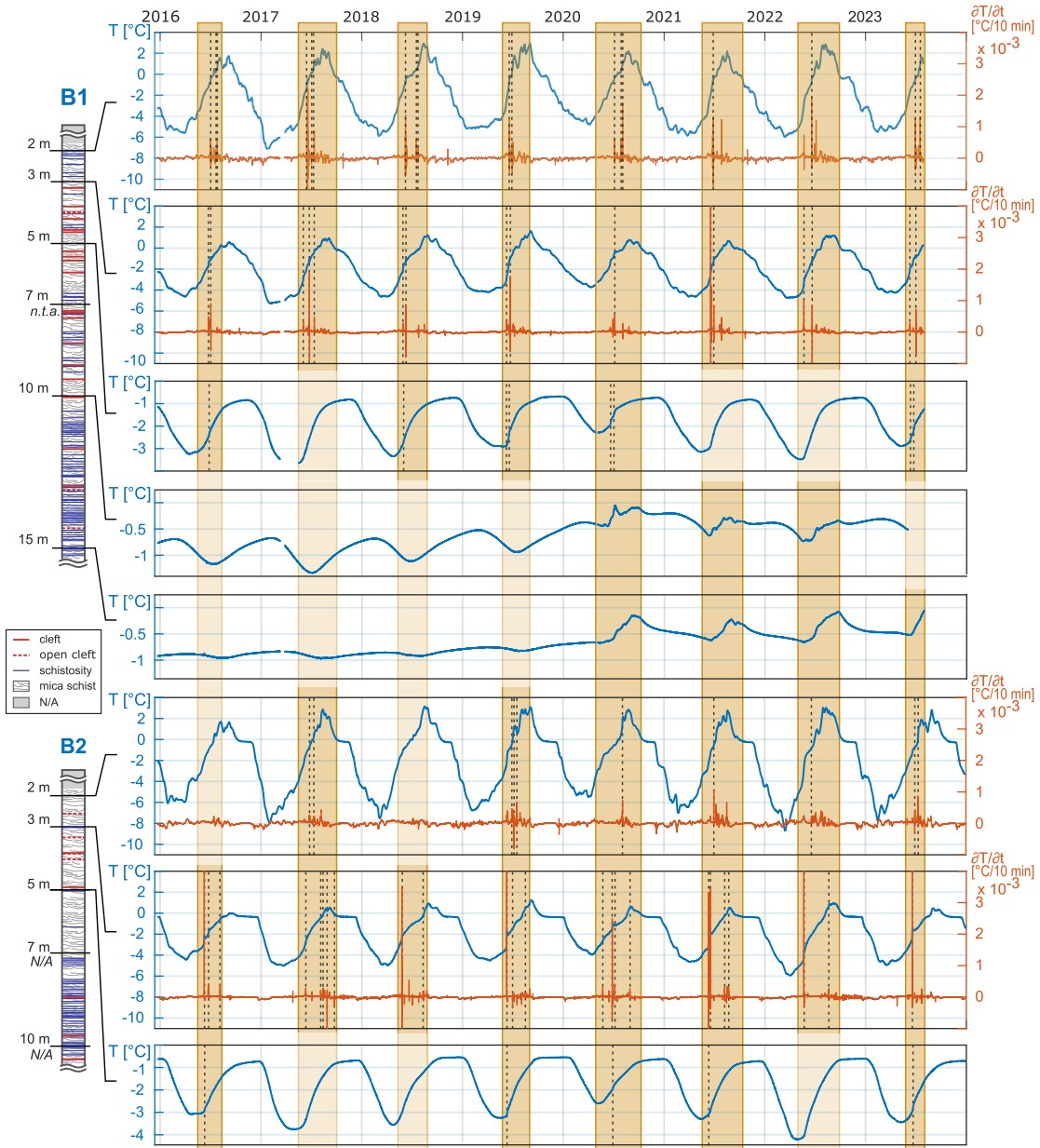

**Figure 5.** Thermal anomalies in rock temperature throughout the observation period (2016-2023): identified at 2 and 3 m depth (B1, B2) by abrupt changes using the first deviation with a moving average of a sample window size of 2 hours, and by irregularities and changes in the quasi-sinusoidal pattern observed in greater depths (B1: 5, 10, 15 m, B2: 5 m). Dashed lines mark an event of thermal disturbance, which mainly occurred between May and September. Thermistors showing no thermal anomalies (n.t.a.) or damage (N/A) are indicated accordingly.

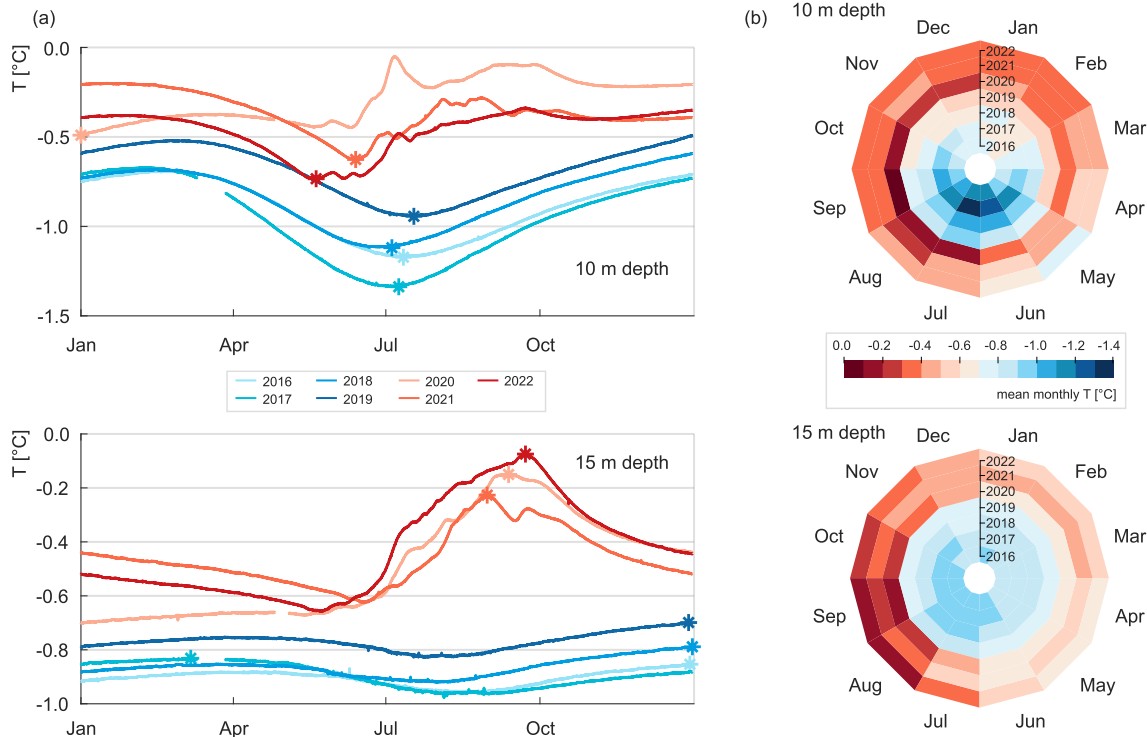

**Figure 6.** Borehole temperature at depths of 10 and 15 m between 2016 and 2022 in B1: a) Thermal signals in 10 m depth, with minimal values highlighted (top) and at 15 m depth, with maximal values highlighted (bottom). b) Mean monthly temperature values, with each ring representing a measurement year and the radius increasing for more recent observation years.

(−3.4 °C). The zero-curtain effect, i.e., an isothermal state where rock temperature stagnates close to 0 °C due to the release of latent heat (Outcalt et al., 1990), was most pronounced at 3 m depth in B2 (Fig. 5).

## 4.2 Seasonal variations in ERT

The seasonal evolution of the ERT acquired from February-June 2013, June-September 2023, and September-December 2013 are shown in Fig. 7. In February, resistivity values greater than 64 kΩm were observed throughout the profile except for a small area of measurement noise (x = 46-48 m). Only slight decreases in resistivity occurred in March 2013 and April 2013 at depths between 5 and 10 m, and over the entire observation depth (0-10 m) in April 2013, June 2013, and June 2023. Nevertheless, electrical resistivity values remained above 32 kΩm, indicating frozen rock and air-/ice-filled joints. Borehole temperature data from B2 (Fig. 4a) suggests an active layer thickness of approximately 1.7 m at the time of the ERT measurement in June 2023, which is in good agreement with near-surface (∼ 0-2 m depth) electrical resistivity values of less than 12 kΩm in the lower part of the tomography (∼ x = 28-58 m). The most pronounced seasonal variation in electrical resistivity occurs in summer (Fig. B1). In July, the low-resistivity zone in the lower part of the tomography expands to ∼ 5 m depth while resistivity drops

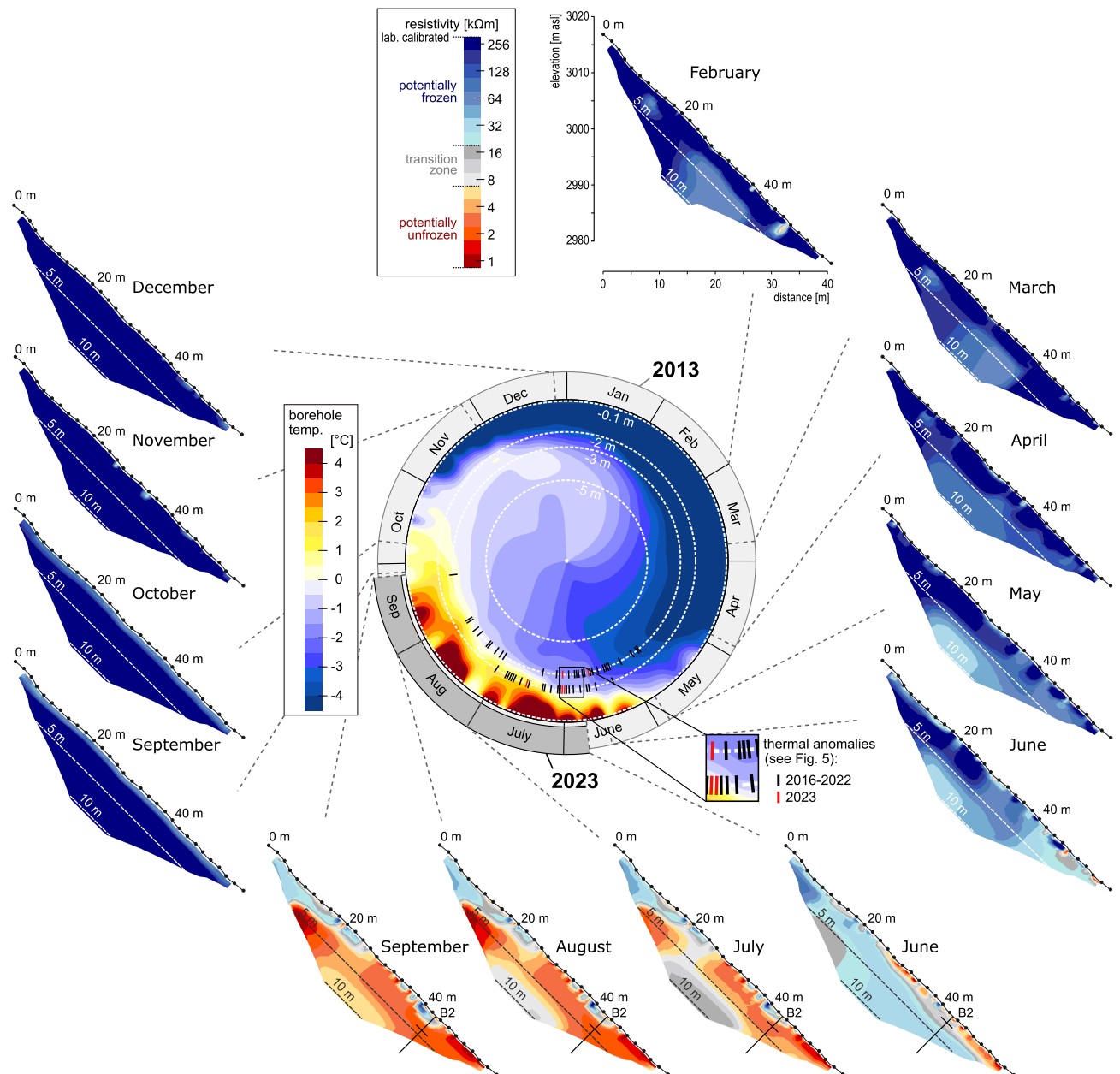

**Figure 7.** Tomographies of repeated ERT measurements from February to June 2013, June to September 2023, and September to December 2013, revealing seasonal resistivity changes. The position of borehole B2 is indicated in the resistivity profiles 2023, with horizontal lines representing the 0°C threshold at the date of the ERT measurements. The central polar plot illustrates rock temperature recorded in borehole B2 from January-December 2023, indicating measurement depths by dashed lines. Thermal anomalies (abrupt temperature changes) found in both boreholes over the entire observation period (2016-2023) are marked at the respective time and depth.

to around 4 kΩm. In July and August, this trend continues as the low-resistivity zone extends to the bottom of the tomography (∼ 10 m depth). The upper part of the profile (∼ x = 0-18 m) is unaffected by the trend. A high resistivity body of ≥ 32 kΩm remains stable during the summer month, probably due to the shielding of water infiltration through the cable car station and consequent desiccation of the surface rock layer, combined with an intact rock mass without major fractures. Coinciding with the freezing period from September to December 2013, high electrical resistivity values reappeared (≥ 128 kΩm).

The general shift towards lower resistivity values from June to September is also clearly discernible in the raw data (Fig. B2). The median values of the apparent resistivities decreased from 15.6 kΩm in June 2023 to 5.5 kΩm in July and remained at low levels from August (4.0 kΩm) to September (3.4 kΩm). Pre-processed data from ERT measurements in June (2013 and 2023) show comparable characteristics with a wide range of apparent resistivity values and a reasonable number of data points above 19 kΩm (i.e., laboratory threshold of frozen conditions). From July to September, apparent resistivities greater than 19 kΩm occurred sporadically at shallow depths, whereas values characteristic for unfrozen conditions (< 10 kΩm) are more pronounced and were observed solely in deep-penetrating measurements.

### 4.3 Temperature-resistivity laboratory calibration

Figure 3b shows the rock temperature related to the measured electrical resistivity of the laboratory analysis. Warming of the frozen rock samples (n = 7) starting at −3 °C followed a linear temperature-resistivity behavior with a steep gradient of −44.9 ± 12.0 kΩm/°C up to the equilibrium melting point between −0.4 and 0.2 °C. The resistivity values at the equilibrium melting point ranged from 10-19 kΩm, which is subsequently used as a threshold for the transition zone between frozen and unfrozen conditions for the inverted tomograms of the Kitzsteinhorn rock face. The unfrozen temperature-resistivity paths exhibit a flatter gradient of −0.6 ± 0.2 kΩm/°C. The fitting parameters and respective $R^2$ of the curves are listed in Table D1.

### 4.4 Piezometric pressure

The piezometric pressure showed a slight increase from 0.21 to 0.49 bar between January and early April in 2024 (Fig. 8). This was followed by intervals of rapid increase during spring and summer: from 0.49 to 0.78 bar in mid-April, from 0.87 to 1.07 bar at the end of May, and from 0.97 to a peak value of 1.18 bar in June. The rapid rise in piezometric pressure correlates with days when the mean air temperature was above 0 °C, as measured at the weather station on the nearby Gletscher Plateau (2.940 m asl). Notable daily fluctuations in the piezometric signal were observed from early to mid-May and in June.

## 5 Discussion

### 5.1 Pressurised water flow in permafrost rockwalls

The unique time series of laboratory-calibrated ERT observations presented in this paper enable a quantitative interpretation of seasonal changes in frozen rockwalls. High-resistivity sections above 19 kΩm indicate frozen conditions with air-/ice-filled joints during the frost season (October-May). Slight warming of the rock surface after the snow cover disappears in late spring

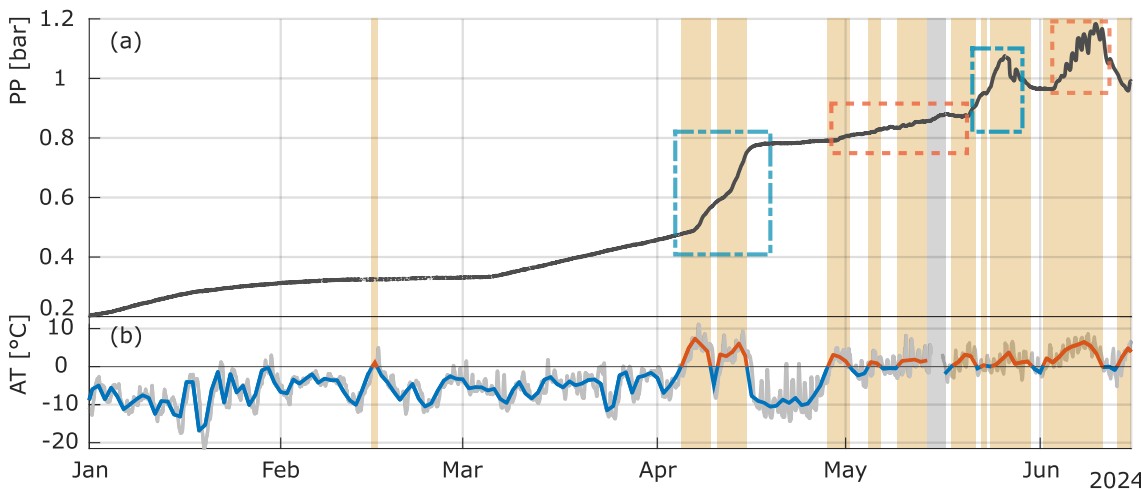

**Figure 8.** a) Piezometer pressure (PP) from January to June 2024 recorded near the summit station at a depth of 16.85 m. b) Mean daily air temperature (AT) from the weather station at the Gletscher Plateau (2.940 m asl, distance $\sim$ 500 m), shown in blue ($< 0$ °C) and red ($> 0$ °C). The moving mean air temperature (AT) over a 2-hour interval is represented in grey. Yellow bars indicate periods when the mean daily air temperature was $> 0$ °C, during which increases (blue rectangle) and short-term fluctuations with 24-hour frequency (orange rectangle) in pressure level were mostly observed. The grey balk marks a data gap in the weather station recordings.

is indicated by decreasing resistivity at shallow depth (e.g., tomography from June 2023 in Fig. 7). Ice-filled joints probably act as an aquitard, constraining deep infiltration into the joint system, with snowmelt mainly draining along the rock surface. From June to July, rapid changes in resistivity of more than one order of magnitude were observed at $\sim 1-7$ m depth coincident

with a borehole temperature warming accompanied by active layer deepening from 1.7 to 2.7 m depth between the ERT measurement dates in June and July (Fig. 9a). The low resistivity zone ($\sim 4$ k$\Omega$m) in July in the lower part of the tomography ($\sim$ x = 28-58 m) gradually expands to higher rock slope sections (Fig. 7) and to the bottom of the ERT profile until September ($\sim 10$ m depth, Fig. 9a), while the 0 °C/$-0.5$ °C isotherm (i.e., permafrost table) changes marginally (Aug: 3.5/4.1 m, Sep: 3.5/4.3 m, Fig. 9a).

The term *'pressurised'* here refers to a piezometric head of a few meters. The rapid resistivity decline observed suggests pressurised nature of water flow in fractures, supported by additional evidence. This evidence comprises visually observed water outflow from fractures (Fig. 10) and first piezometric measurements showing rapidly increasing pressure levels in the thawing season, with piezometric heads reaching up to 11.8 m (Fig. 8). Without assuming pressurised flow, the decline in electrical resistivity from July to September (Fig. 7, 9) would be inconsistent with Archie's law. In thawed conditions, resistivity

decreases for various rock types at a rate of $\sim 2.9 \pm 0.3$ %/°C (Krautblatter, 2009), and according to our laboratory calibrations, by $4.5 \pm 0.3$ %/°C (Fig. 3, Table D1). Thus, a temperature warming from July to September (Fig. 9) in already fully saturated rock with constant porosity would not cause a significant further and rapid electrical resistivity decline. This can only occur if pressurised water flow contributes to additional hydraulic opening of fractures within days to weeks. In addition, the coincident

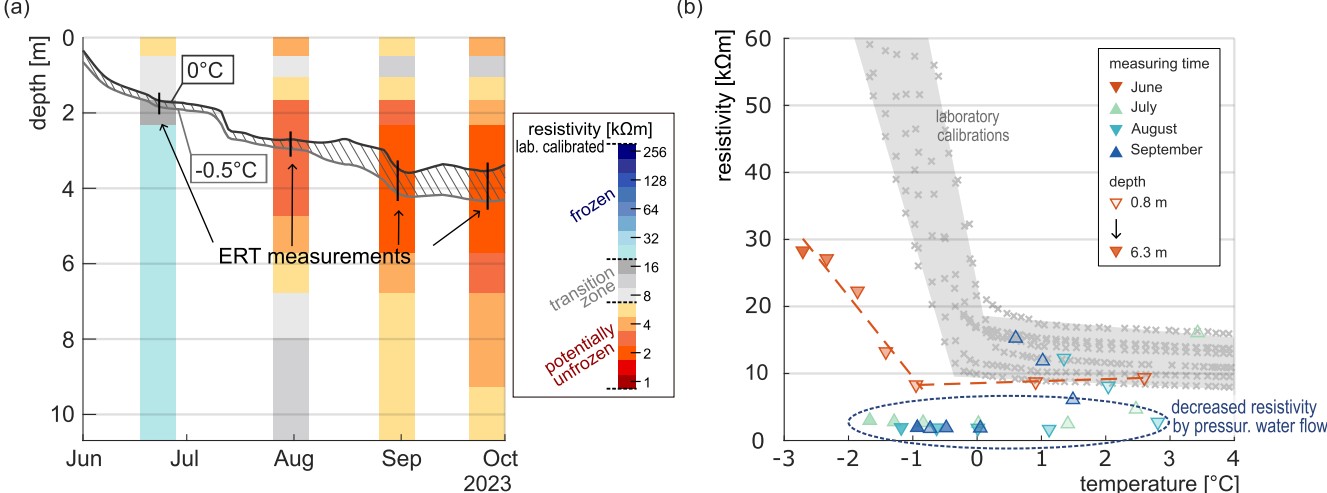

**Figure 9.** (a) Median resistivity values of the ERT measurements from June-September 2023, respectively, over the depth of the inverted blocks and the corresponding temporal evolution of the 0 °C/−0.5 °C observed in B2. Isotherms were calculated using average temperatures at 0.1 m depth over 3 days to minimize short-term fluctuations. (b) The resistivity-temperature relation of the field and laboratory ERT measurements showed a decrease in resistivity values from ∼ −2 to 3 °C obtained between July and September.

rapid changes (Fig. 5) and regime changes in rock temperature (Fig. 6), cannot be explained solely by diffusive heat exchange
(Noetzli et al., 2007; Krautblatter et al., 2010), but only by water flow in open fractures (Phillips et al., 2016), facilitating a
thermal shortcut between the atmosphere and the subsurface (Hasler et al., 2011a).

We hypothesize that water from snowmelt infiltrates in early summer along local joints and fractures. Joint set K2, which dips
steeply to SW (Fig. 1), may play a particularly prominent role. K2 extends perpendicular to the dip direction of the studied rock
slope, effectively collecting surface runoff and diverting it into the subsurface. There, it is most likely redistributed along the
schistosity of the local calcareous mica-schists, i.e., through open, surface-parallel joints that drain the infiltrated water to the
base of the rock slope. This hypothesis provides a plausible explanation for the observed development of a low-resistivity zone
in the lower half of the ERT tomography in July, in which water seeps through one prominent K2 joints with apertures around
5 cm (Fig. 10) and is then drained down-slope and pressurised up-slope along the schistosity in August and September. In
addition to snowmelt, rainfall events can contribute to infiltration, potentially leading to short-term warming in the subsurface
related to non-conductive heat fluxes, and decreased resistivity. No rainfall has been registered in the days leading up to or
during the ERT data acquisitions (Fig. A1), so modifications of the hydrological regime through precipitation are not discussed
here.

The results of this study underline the high impact of fluid flow in fractures on hydrological and thermal processes in alpine
permafrost rockwalls. Due to its complexity and lack of knowledge, thermal models of permafrost distribution have often
neglected the hydrology of rockwalls (e.g., Duvillard et al., 2021a; Etzelmüller et al., 2022), or have relied on an assumption
of weathered zones and high porosity to represent fracture density (e.g., Noetzli et al., 2007; Magnin et al., 2017), which does

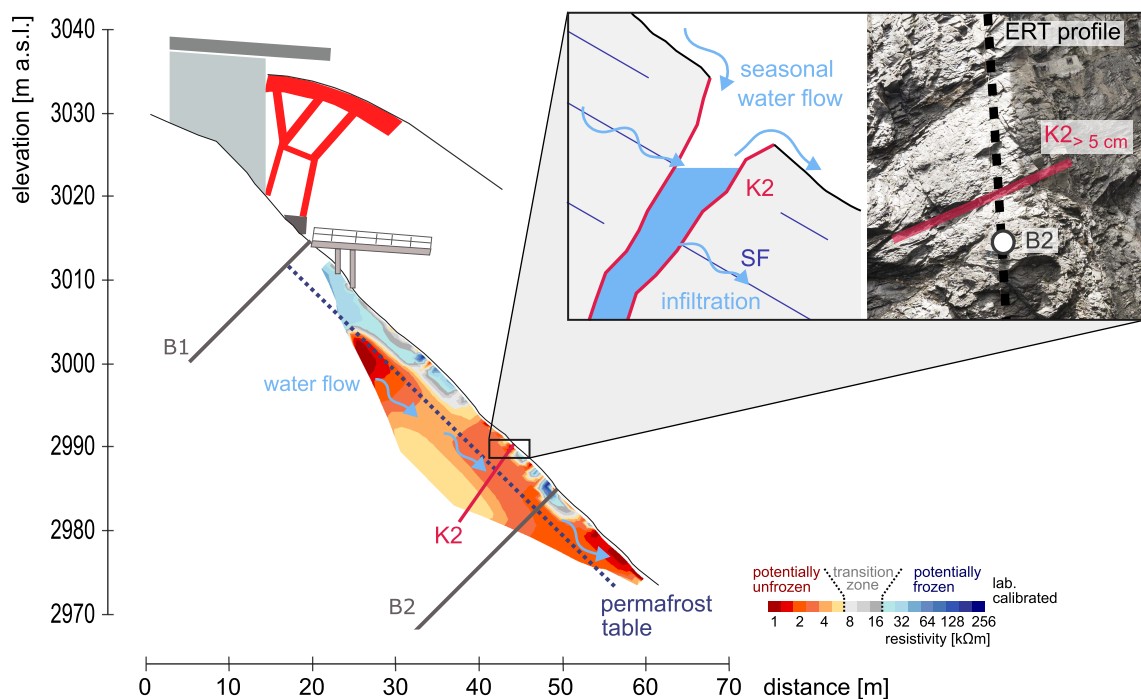

**Figure 10.** Conceptual illustration of water infiltration and flow along the schistosity (SF) and geotechnical discontinuities (e.g., fracture K2 with aperture of > 5 cm) using the ERT measurement from September 2023. Permafrost table is derived from temperature measurements at borehole B2.

not fully describe the main effects of a fracture network on permeability characteristics in low porosity rock (Hsieh, 1998). However, fluid flow in fractures is often decisive for rapid changes in permafrost distribution in rockwalls as already shown by geophysical measurements (Krautblatter and Hauck, 2007; Krautblatter et al., 2010; Keuschnig et al., 2017; Magnin and
Josnin, 2021), numerical approaches (Magnin and Josnin, 2021) and our combined analysis of long-term ERT monitoring and borehole temperatures. As evidenced by several rapid drops in rock temperature recorded at the Kitzsteinhorn and at the Gemsstock borehole in Switzerland (Phillips et al., 2016), we suggest that fractures can act as cooling pathways, favoring the formation of freezing corridors and air ventilation. Ice-filled fractures initially cool the surrounding rock mass and prevent warm water infiltration. Still, as the melting of the basal ice layer begins, thaw corridors are formed, which initiate rapid
non-conductive warming in spring/summer. Cooling in autumn/winter, on the other hand, is controlled by slow conductive heat transport and contributes to the persistence of relatively warm areas at larger depths. Beside the possibility of varying subsurface thermal conductivity (Hasler et al., 2011b), this water-flow-induced seasonal succession of rapid warming and slow cooling can result in a positive thermal offset, as observed in B1 and B2 (Fig. 4a), which, over long time periods, results in bottom-up oriented permafrost degradation.

## 5.2 Limitations and uncertainties

Borehole temperature measurements are vulnerable to lightning strikes, leading to sporadic thermistor failures, as happened in several depths (0.5, 7, 10, 20, 25 m) in B2 between 2017-2023. Therefore, the frequency of thermal anomalies recorded in B2 may be underrepresented. The overall record of thermal anomalies, however, is derived from a seven-year dataset (2016-2023) and in numerous depths (B1: 2, 3, 5, 7, 10, 15 m, B2: 2, 3, 5 m), still holds high significance. The system used for borehole temperature measurement facilitates direct physical contact between sensors and ambient rock mass and is, therefore, ideally suited to record short-term temperature changes. However, despite their high sensitivity, the measured thermal signals are assumed to be considerably damped compared to direct temperature measurements in fractures or cracks. Actual temperature jumps induced by water flow in fractures are expected to be significantly higher than the ones measured inside B1 and B2.

Sources of error in geophysical models are associated with data acquisition and subsequent inversion. Keuschnig et al. (2017) indicated that wind gusts, especially above 10 m/s, might induce measurement errors. Here, we only used ERT measurements from 2013 and 2023, carried out during periods with wind speeds below 10 m/s, making additional pre-filtering superfluous. Unfortunately, the ERT measurements in 2023 could not be repeated with the same instrument as in 2013, which might affect the comparability of the datasets (Table B1). However, the overlapping observations in June showed comparable trends in electrical resistivity towards lower values, particularly in the lower part of the profile, suggesting only a minimal influence of the different instruments. Atmospheric conditions vary slightly between years, affecting the timing of snow melt and hence the change in resistivity regime. The ERT results from June 2013 exhibit higher resistivity values in the upper part of the profile ($> 64$ kΩm) compared to the ERT measurement in 2023 (Fig. 7), probably due to colder rock and atmospheric conditions prior to the measurement (Fig. A1). The penetration depth of the current flow into the subsurface depends on the characteristics of the top layer and may vary seasonally. Dry and frozen conditions can impede current flow, while water-saturated conditions might trap current flow, resulting in an attenuated current flow into deeper layers (Loke, 2022). Poor electrode coupling is often associated with frozen conditions and ice-filled fractures and cracks from autumn to spring, which cause noisy data and can lead to inversion artifacts represented by a high RMS error. This phenomenon was observed, for example, in September 2013 vs 2023, where cold air temperatures likely caused freezing of the rock surface layer prior to the ERT measurement (Fig. A1), resulting in high contact resistance (Table B1) and an inability to resolve the long-lasting summer thermal signal in greater depths. Consequently, we refrained from detailed interpretation of the inverted tomograms from February-May 2013 and September-December 2013 but included the ERT data to cover all season. We assume that a repeat of the ERT measurements in 2023 during the frost period would have yielded comparable results, as borehole temperature show frozen subsurface conditions from January-June and from October-December 2023 (Fig. 7), during which no thermal anomalies or irregularities were observed (Fig. 5). In addition, we analyzed the raw data from June-September 2023 to exclude an inappropriate choice of inversion parameters, even though the tomograms exhibit a marginal RMS error of less than 4.3 %. The apparent resistivity values indicate a decreasing trend over the summer (Fig. B2), consistent with the patterns shown by the respective inverted tomograms.

Laboratory calibrations are commonly applied to enable a quantitative interpretation of ERT monitoring in permafrost bedrock. We carried out laboratory ERT measurements on unfrozen and frozen mica-schist rock samples, representative of the homogeneous lithology of the study side, with only a scaly band of serpentinite at a depth of approximately 15 m according to optical borehole scans. As the ERT measurements at the north flank reached a maximum penetration depth of around 10 m, serpentinite probes were not included in the laboratory experiments. A bilinear description of the resistivity-temperature relation was preferred over an exponential behavior of resistivity values below the freezing point, which is less appropriate for low-porosity bedrock (Krautblatter et al., 2010) compared to freezing in unconfined space (e.g., debris) and requires more unknown input parameters (Duvillard et al., 2021a). Our results suggest frozen conditions above 19 kΩm and a frozen temperature-resistivity gradient of 44.9 ± 12.0 kΩm/°C, comparable to laboratory calibrations of rocks with different lithologies (Krautblatter et al., 2010; Magnin et al., 2015; Scandroglio et al., 2021; Etzelmüller et al., 2022). However, the problem of extrapolating from laboratory experiments to field observations (Zisser et al., 2007; Krautblatter et al., 2010) was highlighted by our ERT observation of the highly fractured rock face with anisotropic characteristics, which indicated the strong influence of water-saturated fractures and cracks on the electrical properties, less represented by the intact rock mass of the laboratory studies.

## 5.3 Implications for rock wall instabilities and permafrost monitoring

Understanding water flow in fractured permafrost rocks is challenging but crucial to improving process knowledge of permafrost dynamics. The interaction between fracture fills of air, water and ice, and the surrounding bedrock results in complex thermal patterns that strongly influence permafrost evolution and have further implications for the mechanical rock regime.

High, seasonally varying displacement rates of rock slope instabilities (Wirz et al., 2014; Mamot et al., 2021; Etzelmüller et al., 2022) and irreversible fracture displacements in summer (Weber et al., 2017) have already been linked to thawing-related processes such as meltwater infiltration in cracks and fractures. Induced hydraulic and/or hydrostatic pressures may induce uplift, which reduces the effective normal stress and thus the shear resistance (Wyllie, 2017), and can contribute as a triggering factor to slope failures in alpine rocks (Fischer et al., 2010; Walter et al., 2020; Kristensen et al., 2021). Our results show that repeated ERT monitoring can detect water infiltration and is, therefore, a promising approach to identify periods with high hydrostatic pressure relevant for slope stability assessment.

Temperature regime changes and abrupt temperature changes recorded in B1 and B2, mainly during periods of subzero temperature (Fig. C1), indicate non-conductive heat fluxes. These sudden push-like events can transport heat from the surface to greater depths more rapidly than slow thermal conduction, which is consistent with laboratory experiments by Hasler et al. (2011a). The induced temperature rise may lead to slope destabilization, as warming and thawing processes alter the shear resistance of ice-filled rock joints (Gruber and Haeberli, 2007; Mamot et al., 2018) and affect the rock-ice mechanics of permafrost slopes (Krautblatter et al., 2013).

# 6 Conclusions

The role of water flow in permafrost rockwalls is rarely investigated and remains a key challenge. This study proposes an approach to infer water flow in bedrock permafrost by combining monthly repeated electrical resistivity monitoring, long-term temperature measurements in deep boreholes, and piezometer observations. The following conclusions are drawn:

1. A massive decrease in electrical resistivity values during the thawing season (July-September) can be indicative for snow melt water infiltration into the rockwall draining along the schistosity and interconnected joints, and subsequently becoming pressurised within a widespread fracture network.

2. Hydrostatic pressure levels of up to 11.8 m indicate a widespread water infiltration into the fracture network, which potentially alters slope stability by favoring bottom-up permafrost degradation.

3. Small, abrupt temperatures anomalies registered in the two boreholes (2, 3 and 5 m depth) suggest non-conductive heat flux in fractures. Frozen rock is warmed more rapidly by these sudden push-like events of heat transport from the surface than by slow thermal conduction alone.

4. Long-term regime temperature changes were identified in two boreholes in 10 and 15 m depth between 2016-2019 and 2020-2022, indicating the pronounced heat transfer by infiltrating water.

5. Monitoring of alpine permafrost often relies solely on annually repeated geoelectric measurements, mainly due to complicated logistics and harsh measurement conditions. However, our study suggests that higher ERT measurement intervals are required to decipher the complexity of hydrothermal processes in permafrost rockwalls and fully assess the rate and extent of permafrost evolution. Monthly repeated measurements in this contribution represent a significant advancement compared to annual surveys.

6. We emphasize the key role of complementary temperature measurements and their joint analysis. Low electrical resistivity values in the absence of borehole temperatures may be misinterpreted as permafrost-free rock slopes, yet. They could serve as an indicator of water-saturated conditions above a potential permafrost body.

This study has broad implications for understanding hydrothermal processes in steep, fractured rock walls, profoundly impacting the rate and extent of permafrost degradation and related hazards. Future developments are needed to validate and quantify our observations. Of particular interest would be simultaneous electrical resistivity and piezometric measurements during the thawing season, whereby daily or hourly observation intervals would represent another significant step towards a better understanding of the transient nature of water flow in fractures.

## Appendix A: Weather conditions during the ERT observation years 2013 and 2023

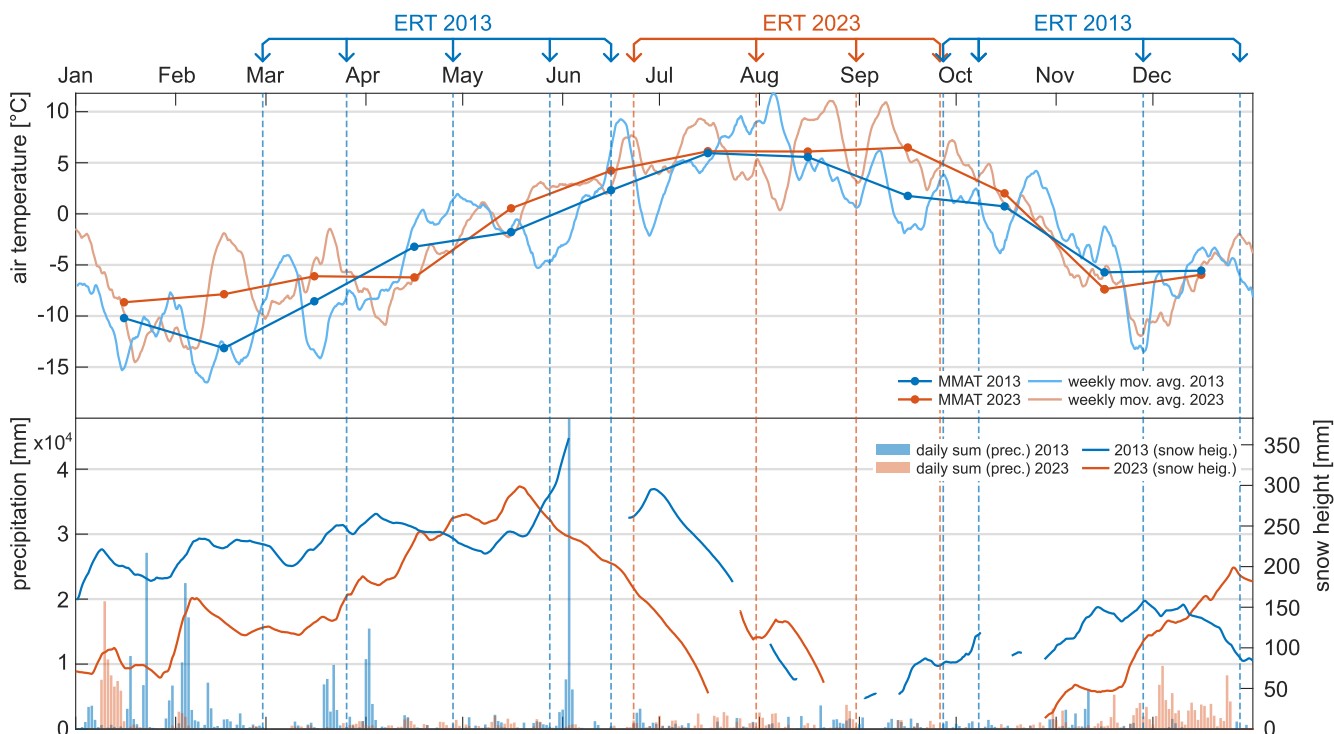

**Figure A1.** Mean monthly air temperature (MMAT) and the weekly moving average for the air temperature for 2013 and 2023 were recorded at the nearby weather station at the Glacier Plateau at 2.940 m asl ($\sim$ 500 m distance from the study site). The MMAT in 2013 and 2023 showed comparable values for most months ($\Delta$ *MAAT* < 2.5 °C), with notable differences in February (2013: -13.1 °C/2023: -7.9 °C), April (2013: -3.2 °C/2023: -6.2 °C), and September (2013: 1.8 °C/2023: 6.5 °C). Snow height was also recorded at the Glacier Plateau weather station. Although the slope angle at the weather station is lower than the surrounding the study site, the orientation is the same, making the snow height trends and subsequent snowmelt infiltration patterns transferable to the study site, albeit not the absolute values. Snow height in 2013 and 2023 is visualized with a solid line, with interruptions due to data gaps. The snowmelt was most pronounced in late May 2013 and in June 2023. The bar plot shows the daily rainfall sum at the weather station at the Alpincenter (2.450 m asl, distance < 2 km). Notably, no heavy rainfall events were registered shortly before or on the days of ERT measurements.

**Appendix B: ERT monitoring: Short-term variations, technical details and inversion parameters, raw data analysis, and difference in resistivity from June-September 2023**

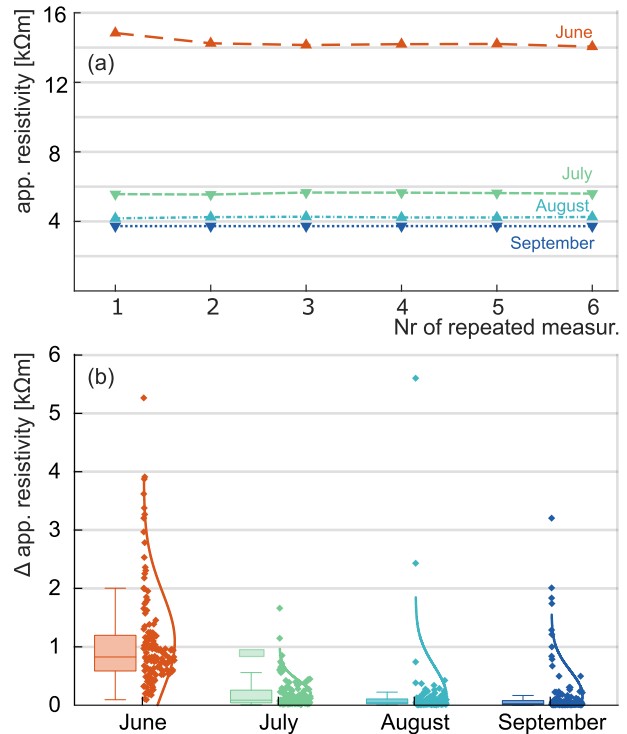

**Figure B1.** Short-term variations in apparent resistivity during repeated ERT measurements (June-September 2023) over a 24-hour period. (a) Median apparent resistivity values indicate relatively stable conditions over an observation period, which is additionally supported by (b) the maximum difference in apparent resistivity of respective datum points (n=135) over repeated ERT measurements, mostly less than 2 kΩm.

**Table B1.** Detailed information on the ERT measurement procedures and inversion routines. ERT measurements were carried out at the end of each month, expect in October due to technical issues. Although different instruments were used in 2013 and 2023, which could slightly impact the comparability of datasets, the overlapping ERT measurements in June showed comparable trends in electrical resistivities towards lower values, suggesting minimal influence from the different devices. During the frost period, contact resistances were determined to > 200 kΩ, whereas during the thawing period, they were within an acceptable range, mostly < 190 kΩ. The filtering of the outliers and inversion routines were standardized across all ERT measurements.

| | 2013 | | | | | 2023 | | | | 2013 | | | |
|---|---|---|---|---|---|---|---|---|---|---|---|---|---|
| | Feb | Mar | Apr | May | June | June | July | Aug | Sep | Sep | Oct | Nov | Dec |
| Date | $28^{th}$ | $26^{th}$ | $28^{th}$ | $28^{th}$ | $16^{th}$ | $23^{th}$ | $31^{st}$ | $31^{st}$ | $26^{th}$ | $27^{th}$ | $08^{th}$ | $28^{th}$ | $28^{th}$ |
| Device | Geo-Tom-MK1E100 | | | | | ABEM terrameter LS | | | | Geo-Tom-MK1E100 | | | |
| Output current [mA] | 0.001 - 200 | | | | | 0.1 - 2500 | | | | 0.001 - 200 | | | |
| Voltage max [V] | 500 | | | | | 600 | | | | 500 | | | |
| Filtering current effects [Hz] | 16 2/3, 50, 60 | | | | | 50 | | | | 16 2/3, 50, 60 | | | |
| Stacks | 3 | | | | | 2-4 | | | | 3 | | | |
| Contact resistance [kΩ] | >200 | | | | - | <150 | <190* | <190 | <170 | >200 | | | |
| No. of outliers/ | 2/ | 1/ | 1/ | 1/ | 1/ | -/ | -/ | -/ | -/ | -/ | -/ | 3/ | 1/ |
| inverted datum points | 133 | 134 | 134 | 134 | 134 | 135 | 135 | 135 | 135 | 135 | 135 | 132 | 134 |
| Iterations | 5 | 5 | 5 | 5 | 5 | 5 | 5 | 5 | 5 | 4 | 5 | 4 | 3 |
| RMS error [%] | 53.0 | 32.7 | 18.9 | 12.7 | 12.0 | 4.3 | 3.6 | 3.8 | 4.2 | 30.7 | 2.1 | 55.2 | 53.2 |

*one measurement indicated contact resistance of 238 kΩ

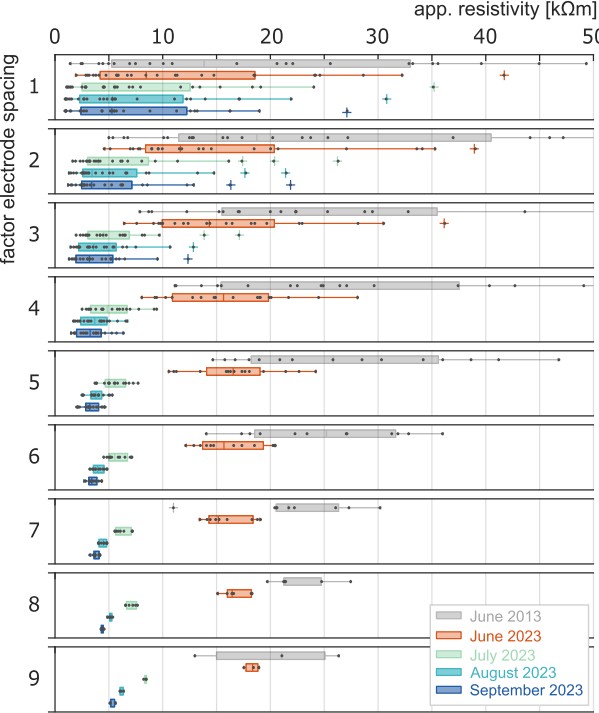

**Figure B2.** Statistical box plots with jitter data points of the apparent resistivity of the ERT measurements in June 2013 and from June-September 2023 with respect to the distance between the electrodes. It should be noted that as the electrode spacing factor increases, fewer values are measured. The electrical resistivity values indicated a general shift towards lower resistivity values from June-September.

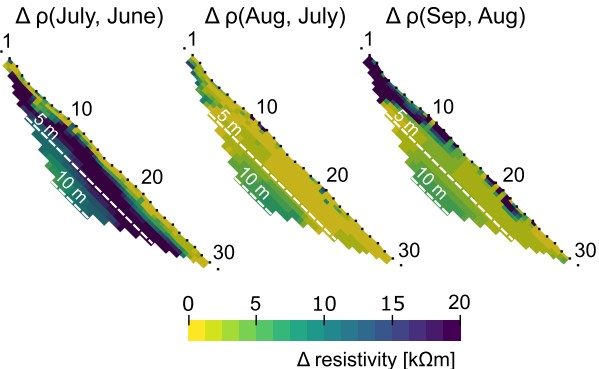

**Figure B3.** Difference in resistivity between tomograms observed from June-July (left), July-August (middle), and August-September (right) in 2023. Significant decreases in electrical resistivity values are evident at greater depths (∼2-7 m) between June-July, while resistivity values remain relatively stable with only slight changes (< 10 kΩm) between July-September.

## Appendix C: Abrupt changes in borehole temperature

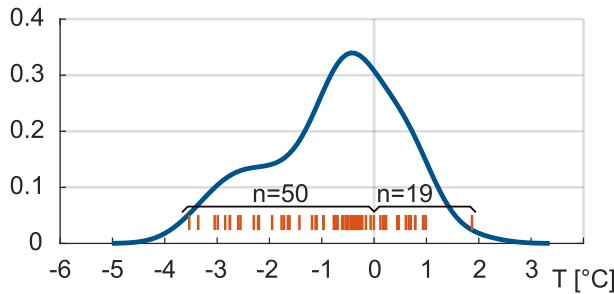

**Figure C1.** Kernel distribution of rock temperature at 2 and 3 m depth (B1, B2) prior to registered thermal anomalies: Most rapid warming or cooling events (anomalies) occurred at subzero temperature.

## Appendix D: Resistivity-temperature laboratory calibration

**Table D1.** Fitting parameters of the bilinear relationship ($a+b*T$) between resistivity and temperature data below and above the equilibrium freezing point $T_f$ of the laboratory experiments.

|  | $T_f$ | below $T_f$ | | | above $T_f$ | | |
|---|---|---|---|---|---|---|---|
|  |  | $a$ | $b$ | $R^2$ | $a$ | $b$ | $R^2$ |
| S1-1 | 0.22 | 23.6 | -48.7 | 0.99 | 13.2 | -0.6 | 0.77 |
| S1-2 | -0.24 | 2.9 | -56.9 | 0.99 | 16.3 | -0.7 | 0.93 |
| S2-1 | -0.35 | -2.0 | -32.8 | 0.99 | 9.6 | -0.4 | 0.91 |
| S2-2 | -0.26 | 7.7 | -34.6 | 1.00 | 16.5 | -0.8 | 0.83 |
| S3-1 | -0.31 | 1.0 | -33.6 | 1.00 | 11.3 | -0.5 | 0.79 |
| S3-2 | -0.06 | 13.1 | -35.3 | 1.00 | 15.3 | -0.7 | 0.82 |
| S3-3 | 0.07 | 22.6 | -48.7 | 1.00 | 19.0 | -0.8 | 0.76 |

*Data availability.* Data will be made available on request.

*Author contributions.* MO wrote the manuscript, conducted the geophysical surveys in 2023, performed the data analysis and designed the figures with substantial contribution from SW and MKe. MKe provided the data sets of geophysical surveys in 2013 and IH contributed borehole temperature and piezometer data. SW, MKr and MKe supervised the study. All authors contributed to the interpretation of the
results and improved the final version of the manuscript.

*Competing interests.* The authors declare that they have no conflict of interest.

*Acknowledgements.* Maike Offer acknowledges PhD funding from the Deutsche Bundesstiftung Umwelt (DBU). The authors would like to thank Maximilian Rau, Felix Pfluger, Georg Stockinger, and Maximilian Reinhard for their extensive support during the fieldwork. We furthermore thank the Gletscherbahnen Kaprun AG (Project 'Open-Air-Lab Kitzsteinhorn') for financial and logistical support. We are
grateful to the editor, Teddi Herring, and two anonymous reviewers for their constructive comments and suggestions.

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
