# Peer review of "Pressurised water flow in fractured permafrost rocks revealed by borehole temperature, electrical resistivity tomography, and piezometric pressure"

_EGUsphere, 2024_

## Author Comment (AC1)

**Author comment:**

Dear Victor Pozsgay,

Thank you very much for your community comment on our manuscript titled *"Pressurised water flow in fractured permafrost rocks revealed by joint electrical resistivity monitoring and borehole temperature analysis" (egusphere-2024-893).* We sincerely appreciate your constructive and positive feedback. We have provided general responses to your community comments in blue. However, we encourage you to also review our detailed responses to the reviewer comments, as some of your concerns are addressed there more thoroughly.

With kind regards,

Maike Offer, Samuel Weber, Michael Krautblatter, Ingo Hartmeyer, and Markus Keuschnig

I am a postdoctoral fellow with growing expertise in numerical simulations of permafrost ground in mountain areas and slope failures. I have a background in theoretical physics and am relatively new to the field hence why this comment focuses mainly on scientific methods and data selection, and should be taken with a pinch of salt. It has been a pleasure to read this manuscript and I hope that sharing the following comments will be useful.

The abstract and introduction convey well-written and well-referenced information allowing the reader to understand the context and the interest of the study. However, I believe that the overall scientific methodology could be improved. For instance, conclusions are reached about the timing of the infiltration relative to snow melt and the absence of precipitation in the 'days' preceding measurements but no attempts to consistently measure snow cover or precipitation were made. The influence of the air temperature on snowmelt and on the whole infiltration process is also essential, but once again, the reader does not have access to it. Towards the end, the authors briefly assert that piezometric measurements were made and supported their hypothesis, but neither the method nor the results are reported. In my opinion, the manuscript would be stronger if more supporting evidence was presented to the reader.

Thanks again for your feedback. In the revised manuscript we will provide a weather analysis during the observation period from 2013-2023 (see response A4 for Referee 2). In addition, we will demonstrate the piezometric measurements and describe its methodology and results in detail. Please consider our detailed response to Referee 1 (A2) and 2 (A13) about the piezometric observations.

Beyond this, the major issue that I have with this manuscript lies in the ERT dataset selection. Due to some lightning strikes, most ERT measurements between June and September 2013 were corrupted and the authors decided to fill the gap with data from 2023, 10 years after the original measurements. The authors are comparing monthly ERT measurements coming from two sets of measurements spaced by 10 years, and do not address the issues created by such a significant temporal gap. As they correctly put it in their introduction, the rise of temperatures and the permafrost degradation have accelerated in the last decade, and there is little reason to believe that the study site has not been affected too. In fact, it is clear from Figure 4 that the resistivity of the bedrock along the survey line has changed tremendously between 2013 and 2023 during the months of June and September (the only months measured both in 2013 and 2023). Visually, the most impressive difference comes from September where the resistivity of the whole cross section is about 2 orders of magnitude smaller in 2023 than in 2013. Finally, when looking at the measurement dates in Table B1, I find it surprising that most are taken within the last week of the month but for some unknown reason (which could be technical, but it is not communicated), the October 2013 data was measured on the 8th, which is not

consistent with other data points. Given these comments, it is hard to justify treating the 2013 and 2023 months on an equal basis which is why I believe that the authors could improve the overall readability by sharing their reasoning behind choosing this particular dataset. It would be interesting to know if they are aiming at studying inter-annual or solely seasonal variability, in which case they would probably need to justify why they look at data taken 10 years apart. However, having such data could still be a strength if more was said about the evolution of some metrics over this decade.

The selection of the ERT dataset will be explained more explicit in the revised manuscript. As this was also addressed by Referee 2, we include here our response (A12):

*Since the novelty of our manuscript is to reveal pressurised water flow during the thawing period, we refrained from a detail interpretation of the decadal permafrost change. Our initial motivation to include the data from 2013 was to cover all seasons, including the frost period, which was logistically not feasible in 2023, but would probably have yielded comparable results. We will include a short paragraph in the discussion section 5.3 to clarify our choice of ERT data and suggest to refraining from a detailed interpretation of the decadal permafrost change, since we did not present ERT measurements from the same seasonal periods nor a decadal recording of borehole temperature or piezometric data (Figure R1):*

(Line 324): *"[…] Consequently, we refrained from detailed interpretation of the inverted tomograms from February to May 2013 and September to December **2013 but included the ERT data to cover all season. We assume that a repeat of the ERT measurements in 2023 during the frost period would have yielded comparable results, as borehole temperature show frozen subsurface conditions from January to June and from October to December (Figure 4), during which no thermal anomalies or irregularities were observed (Figure 6).**"*

The differences in the ERT results from September in 2013 and 2023 is likely driven by varying atmospheric conditions before the respective measurements. This was also already responded thoroughly in our response to Referee 2 (A12):

Line 320): *"**Atmospheric conditions vary slightly between years, affecting the timing of snow melt and hence the change in resistivity regime. The ERT results from June 2013 exhibit higher resistivity values in the upper part of the profile (>64 kΩm) compared to the ERT measurement in 2023 (Figure 4), probably due to colder rock and atmospheric conditions prior to the measurement. However, the trend towards lower resistivity values, particularly in the lower part of the profile, is also clearly visible in the tomogram from June 2013.** The penetration depth of the current flow into the subsurface depends on the characteristics of the top layer and may vary seasonally. Dry and frozen conditions can impede current flow, while water-saturated conditions might trap current flow, resulting in an attenuated current flow into deeper layers (Loke, 2022). Poor electrode coupling is often associated with frozen conditions and ice-filled fractures and cracks from autumn to spring, which cause noisy data and can lead to inversion artifacts represented by a high RMS error. **This phenomenon was observed, for example, in September 2013 vs. 2023, where cold air temperatures likely caused freezing of the rock surface layer prior to the ERT measurement, resulting in high contact resistance and an inability to resolve the long-lasting summer thermal signal in greater depths.**"*

The decision to include the ERT measurement from October 8[th] was based on technical reasons. However, we believe that ERT observations at the end of October would have yield comparable results. This assumption based on the marginal differences observed in the ERT tomograms from September to December 2013 (Figure 4). Since we refrained from a detailed interpretation of the ERT measurements during winter due to the potentially reduced data quality (see Line 157-160), we did not further scrutinize the slight variation in observation dates during the frozen season.

Finally, the strength of combining the ERT measurements with borehole temperature data is precisely to be able to produce a plot like Figure 7b, providing some elements of proof of the presence of pressurized water flow. To me, this is the main message of the paper, and I believe it goes slightly unnoticed in the current layout. I would suggest emphasizing this result and providing more explanation of the processes at hand and the reasoning underpinning the conclusion.

The primary message of our manuscript is the detection of pressurised fluid flow during the thawing season. To enhance this point, we will incorporate new piezometric measurements, which will underpin our findings of the borehole temperature and ERT analysis. Additionally, we will slightly modify the conclusion (see A15 in the response to Referee 2) to underscore our main message of the paper.

Overall comments on Figures and Tables:

- The axis labels are not centered, and not capitalized.
  Since the journal does not specify guidelines for axis label formatting, we have chosen our own layout.

- The Tables include some repetitions in the units, some confusing symbols, and some labels not previously introduced.
  We have carefully reviewed the Tables and will remove the repetition of the unit in Table B1 (500**V**) and explain the parameter of the bilinear relation in Table A1 (**a+by**). However, we could not identify confusing symbols as we used only common declarations of parameters.

- Some text should accompany the Figures and Tables of the Appendix.
  The Figures and Tables in the Appendix are either explained in detail within the corresponding paragraphs of the main text or have their findings described directly in the Appendix.

- Not all Figures are referenced, and the order of the Figures in the Appendix does not represent their reference order from the main text.
  We will include the reference to Figure B3 in the main text and modify the order of the Figures in the Appendix to place Table B1 before Table A1.

Some extra comments:

- Line 100: Is there a particular reason behind this choice of diameter? Could you comment on how the relation could potentially change with a different diameter?
  The choice of diameter was based on technical constrains imposed by the drill bits used for extracting the core samples. However, we assume that using a different diameter of the core samples for the laboratory calibrations would not have yielded to significant different temperature-resistivity relationships.

- Line 143: What about the weather conditions in 2013? I believe it would be interesting to present some weather data in a table, say more about the air temperature, talk about precipitation, snow etc.
  See comment above.

- Line 163: The ERT doesn't give any information at depth below x = 0m, so could you please clarify why you decided to place B1 at the beginning of the survey line? A short sentence motivating the geometry of the survey would be interesting for the reader.

  The initial placement of boreholes was primarily intended for monitoring rock temperatures and was determined independently from the ERT monitoring. However, since both boreholes were drilled along the profile, we used them to describe the temperature regime of the investigated rock wall and, specifically, borehole B2 to directly link the resistivity values with rock temperature.

- Line 191 / Paragraph 4.2: In relation to previous comments, it might be interesting and even needed to add a paragraph studying the inter-annual variations.
  See comment above.

- Line 232: From Fig. 6, it seems to me that thermal anomalies are identified with thermal rate of change as low as $10^{-3}$ °C/10min. This corresponds to a difference of $1.2 \times 10^{-2}$ °C over an averaging window of 2h, which is an order of magnitude less than the claimed threshold of ~0.2°C over that same period above which heat transfer becomes non-conductive. Could you please provide more information here and clarify the agreement between the Figure and these statements?
  We will modify the sentence to: *"[…] which exhibit a temperature rise of **up to 0.7 °C in less than 2 hours".***

- Figure 6 / B1 / 15m: It is mentioned that there are 'notable changes in the quasi-sinusoidal pattern since 2020' but I believe the reader would benefit from an explanation of the underlying cause for such a change.
  Please consider the detailed response to Referee 2 (A13).

- Line 273: It is surprising to read this sentence about the piezoemetric measurements without context. I would kindly suggest that the authors add some context and most importantly, present some data.
  See comment above and consider our detailed response to Referee 1 (A2) and 2 (A13) about the piezometric observations.

---

## Author Comment (AC2)

**Author comment:**

Dear Referee #1,

Thank you for taking the time to review our submitted manuscript entitled *"Pressurised water flow in fractured permafrost rocks revealed by joint electrical resistivity monitoring and borehole temperature analysis" (egusphere-2024-893).* We greatly appreciate your thorough, constructive and positive feedback, which we have carefully considered. Our detailed point-by-point responses are given below, highlighted in blue, with proposed changes in the revised manuscript indicated in bold. We believe that these revisions and explanations in our responses fully address your concerns and thereby improve the quality and clarity of our manuscript.

Sincerely,

Maike Offer, Samuel Weber, Michael Krautblatter, Ingo Hartmeyer, and Markus Keuschnig

This reviewer has expertise in permafrost field observations, numerical modelling of frozen soil, and (to a lesser degree) permafrost geophysics.

This manuscript presents a unique dataset of repeated ERT, borehole temperature observations, and site characterization in steep permafrost rock. The combined dataset is beautifully presented and affords insights into the evolution of frozen, thawed, and wet zones in the rock. The careful design of temperature observations allowed detecting fast thermal events at depth that are attributed to water infiltration. These are important topics for research in the context of better understanding permafrost moderated climate control on rock instability.

(A1) Thank you for appreciating the unique, long-term data sets which were achieved under challenging field work conditions as well as the careful design of the results.

The manuscript did not convince me that the data revealed pressurized water as stated in the title. The authors support this inference by mentioning piezometric measurements from late summer 2023 (which are not shown or referenced) and the assumption (which is not developed in detail) that pressurised water flow explains the observed rapid electrical resistivity decline. While I am enthusiastic about the data and many of the analyses presented, a clearer focus, structure, and methodology are required for publication. I recommend encouraging resubmission of this manuscript after adjusting focus and conceptual clarity.

(A2) We value your insightful feedback. Initially, we chose not to include piezometric measurements as the installation was completed in late September 2023, which did not cover the periods of the presented borehole temperature data (01/2016-09/2023) and electrical resistivity measurements (02-06/2013, 09-12/2013, 06-09/2023). In addition, the key seasonal period of snow melt was not covered at the time of our initial submission of the manuscript in March 2023.

However, in response to your suggestion, we will now include the data set of one piezometer (depth: 16.85 m) from January to June 2024 in the revised manuscript. Therefore, we will prepare new subchapters describing the methodology and results. The new designed Figure R1 will be included in the revised manuscript, demonstrating an increase in piezometric pressure levels from spring to summer, with maximum heads reaching already up to 11.8 m. These direct observations of water pressure levels strongly support our hypothesis, inferred previously from the electrical resistivity data,

that the rock matrix is influenced by pressurized water which is most pronounced in the season of snow melt (i.e. days with average air temperature above 0°C).

[Figure]

**Figure R1.** a) Piezometer pressure (PP) from January to June 2024 recorded near the summit station at a depth of 16.85 m. b) Mean daily air temperature (AT) from the weather station at the Gletscher Plateau (2.940 m asl, distance ~500 m), shown in blue (<0°C) and red (>0°C). The moving mean air temperature (AT) over a 2-hour interval is represented in grey. Yellow bars indicate periods when the mean daily air temperature was above 0°C, during which increases (blue rectangle) and short-term fluctuations with 24-hour frequency (orange rectangle) in pressure level were mostly observed. The grey balk marks a data gap in the weather station recordings.

Regarding your addressed need for a more detailed explanation of how pressurized water flow is revealed from the observed electrical resistivity decline, we will elaborate on this in the methodology (*3.1 Laboratory calibration of temperature-resistivity relation*) describing the physical principles of Archie's Law and how we concluded from this to the presence of pressurized water in the discussion section (*5.1 Pressurised water flow in permafrost rockwalls*) to improve the overall conceptual clarity:

**3.1 Laboratory calibration of temperature-resistivity relation**

[revised manuscript text omitted]

*We hypothesize that […]"*

Since we present now the piezometric data, we would suggest changing the title of the manuscript to: "***Pressurised water flow in fractured permafrost rocks revealed by electrical resistivity monitoring, borehole temperature and piezometers***".

To achieve a clear structure of the manuscript, the subchapter in methodology and results will be restructured to begin with the borehole temperature data, followed by the electrical resistivity observations and laboratory calibrations, and concluding with the piezometric measurements.

Krautblatter, M. (2009) Detection and quantification of permafrost change in alpine rock walls and implications for rock instability. Ph.D. Thesis. Friedrich-Wilhelms University Bonn.

- ▪ Water flow in fractured permafrost rock has been investigated, and detected with ERT, previously. This study adds to the body of knowledge incrementally. Confident detection of pressurized flow would indeed make it a novel and significant contribution. A more detailed analysis of the thermally detected flow events could likewise be interesting.

(A3) While a more detailed analysis of thermally detected flow events would be indeed interesting, we propose to focus on the novel and significant contribution of the detection of pressurised water flow to maintain the scope and coherence of the manuscript. Including extensive analysis of heat transfer or energy balance would dilute this focus and increase the length of the manuscript, which would be not consistent with your previous and last comment.

- The specific objectives of the research are not clearly articulated. Consequently, the exact state of the art is unclear, the approach and methods cannot be judged in their appropriateness, and the conclusions are not as compellingly underpinned by the evidence presented as they could be.

(A4) To address your concerns, we will introduce the concept of water infiltration in mountain rock slopes in the introduction and explicitly highlight the research gap to clearly articulate the specific objectives of our study. Additionally, we will include new piezometer data (as mentioned in A2) and provide a more detailed analysis of the change of the rock temperature regime at depths of 10 and 15 m between 2016-2019 and 2020-22 in borehole B1 (see A9). These additional analyses will serve as further indicators, alongside electrical resistivity monitoring, of pressurised water flow and will help to compellingly underpin our conclusions.

- Line 300: The cause of the thermal offset stated appears to be speculation. It can equally be explained by transient effects or lateral variation of surface temperature – and neither require invoking thermal effects of water flow.

(A5) We will revise the text to reflect the possible sources of thermal offsets:

(Line 300): "*Beside the possibility of varying subsurface thermal conductivity (Hasler et al., 2011), this water-flow-induced seasonal succession of rapid warming and slow cooling can result in a positive thermal offset, as observed in B1 and B2 (Fig. 5a), which, over long time periods, results in bottom-up oriented permafrost degradation.*"

Hasler, A., Gruber, S., and Haeberli, W.: Temperature variability and offset in steep alpine rock and ice faces, The Cryosphere, 5, 977–988, https://doi.org/10.5194/tc-5-977-2011, 2011.

- Figures 5 and 6: There are strong temporal trends that would provide important background information. Consider showing a depth profile of temporal trends in mean (and maximum?) temperatures over the entire measurement duration. This will help contextualize Figure 4 and its decadal gap.

(A6) As mentioned in Line 162-163 borehole temperature measurements are available only since December 2015. Therefore, it is not possible to show a temporal trend over the decadal gap between the electrical resistivity observations (2013-2023). Within the available data constrains, we have chosen to focus on the thermal regime in 2023 (Figure 4 and Figure 5) to provide a detailed information of the rock temperature during the electrical resistivity measurements in 2023.

- Line 313: The authors state that obstacles exist for interpreting ERT profiles in fractured rock masses based on laboratory measurements on intact samples (an error). Section 5.1 argues that the differences between lab and field point to pressurized water flow (a signal). Can error and signal be distinguished with sufficient confidence? Explain how.

(A7) We are aware of the upscaling effect between the electrical resistivity measurements on intact rock sample and on the fractured rockwall at the Kitzsteinhorn, as mentioned in Line 313. However, we do not consider the resulting differences between laboratory and field observations as errors. Instead, we interpret them as indicator of conditions at the study site that were not or could not be replicated in the laboratory experiments (i.e. rock masses with water-filled fractures). The full argumentation of how we inferred pressurised water flow from the differences between lab and filed measurements is developed in (A2) and will be included in the revised manuscript in section 5.1.

- Line 21: Is permafrost thaw a hazard?

(A10) We will change it to: *"[…] including **rock slope failures from warming permafrost rocks.**"*

- Section 4.2: What is the impact of using summer ERT that has been measured ten years after the profiles in other seasons? Are we interpreting the influence of seasons or a decade of atmospheric warming (see Figure 6)? This needs to be addressed clearly.

(A9) As previously mentioned, the novelty and significant contribution of our study lie in the detection of pressurised water flow and seasonal changes. Consequently, we have focused on the results during the thawing season and refrained from a detailed interpretation of decadal permafrost changes. Logistical constraints made it unfeasible to repeat the ERT measurements in 2023, which is why we included the ERT data from 2013. In addition, the ERT measurements from 2023 show that the signal is influenced by the water content of the rockwall, greatly affecting the detection of permafrost and, thus, making it impracticable to draw conclusions about its evolution. We will clarify our interpretation by modifying the corresponding sentences in the discussion (section *5.2 Limitations and uncertainties*) and include an explanation of the differences in the results from June/September in 2013 and 2023:

(Line 320): *"**Atmospheric conditions vary slightly between years, affecting the timing of snow melt and hence the change in resistivity regime. The ERT results from June 2013 exhibit higher resistivity values in the upper part of the profile (>64 kΩm) compared to the ERT measurement in 2023 (Figure 4), probably due to colder rock and atmospheric conditions prior to the measurement. However, the trend towards lower resistivity values, particularly in the lower part of the profile, is also clearly visible in the tomogram from June 2013.** The penetration depth of the current flow into the subsurface depends on the characteristics of the top layer and may vary seasonally. Dry and frozen conditions can impede current flow, while water-saturated conditions might trap current flow, resulting in an attenuated current flow into deeper layers (Loke, 2022). Poor electrode coupling is often associated with frozen conditions and ice-filled fractures and cracks from autumn to spring, which cause noisy data and can lead to inversion artifacts represented by a high RMS error. **This phenomenon was observed, for example, in September 2013 vs. 2023, where cold air temperatures likely caused freezing of the rock surface layer prior to the ERT measurement, resulting in high contact resistance and an inability to resolve the long-lasting summer thermal signal in greater depths.**"*

(Line 324): *"[…] Consequently, we refrained from detailed interpretation of the inverted tomograms from February to May 2013 and September to December **2013 but included the ERT data to cover all season. We assume that a repeat of the ERT measurements in 2023 during the frost period would have yielded comparable results, as borehole temperature show frozen subsurface conditions from January to June and from October to December (Figure 4), during which no thermal anomalies or irregularities were observed (Figure 6).**"*

- Section 5.3: Some of the statements seem rather confident. They could be shortened and made specific to well supported conclusion and, as such, added as a short outlook paragraph to the conclusion. Some of the other text in the section is better suited for the introduction of a paper.

(A10) We decided to discuss our findings in detail in section 5.3, as we believe they have significant implications for rock wall instabilities and, more generally, for high alpine permafrost monitoring routines. We have already focused on specific processes and ensured that our statements are well-supported by literature and/ or our study. However, we will follow your suggestions and integrating some of the statements from section 5.3 into the conclusion:

Line (377):

" […]

4. *Monitoring of alpine permafrost often relies solely on annually repeated geoelectric measurements, mainly due to complicated logistics and harsh measurement conditions. However, our study suggests that higher ERT measurement intervals are required to decipher the complexity of hydrothermal processes in permafrost rockwalls and fully assess the rate and extent of permafrost evolution. Monthly repeated measurements in this contribution represent a significant advancement compared to annual surveys.*
5. *We emphasize the key role of complementary temperature measurements and their joint analysis. Low electrical resistivity values in the absence of borehole temperatures may be misinterpreted as permafrost-free rock slopes, yet. They could serve as an indicator of water-saturated conditions above a potential permafrost body.*

*This study has broad implications for understanding hydrothermal processes in steep, fractured rock walls, the rate and extent of permafrost degradation, and related hazards.* **Future developments are needed to validate and quantify our observations. Of particular interest would be simultaneous electrical resistivity and piezometric measurements during the thawing season, whereby daily or hourly observation intervals would represent another significant step towards a better understanding of the transient nature of water flow in fractures."**

- The manuscript text should be shortened and edited for clarity in structure and arguments. Some of the referencing could be tightened, giving preference to one good reference backing up a particular argument instead of listing a handful of publications.

(A11) As mentioned in (A2), we will reorder the subchapters in the methodology and results to improve the clarity and structure. These revisions, along with the other modifications, will significantly strengthen our arguments, particularly through the newly designed Figures R1 and R2 (see author comments for referee 2) and the newly included results from piezometric measurements.

---

## Author Response (AR1)

Technical University
of Munich

Technical University of Munich · Chair of Landslide Research
Arcisstr. 21 · 80333 Munich · Germany

The Cryosphere
**Dr. Teddi Herring**

[Figure]

TUM School of Engineering
and Design
Chair of Landslide Research

Maike Offer
Phone: +49 89 289 25867
**maike.offer@tum.de**

Munich, August 19, 2024

**Author's response (egusphere-2024-893)**

Dear Dr. Teddi Herring,

Thank you very much for the positive evaluation of our submitted manuscript (egusphere-2024-893) entitled "*Pressurised water flow in fractured permafrost rocks revealed by joint electrical resistivity monitoring and borehole temperature analysis*".

We have carefully considered your concerns and those raised by two reviewers and one community commenter. In response to your suggestion to provide more fundamental evidence for our conclusion, we have decided to include piezometric pressure data in the revised manuscript. Therefore, we propose to change the title to "*Pressurised water flow in fractured permafrost rocks revealed by borehole temperature, electrical resistivity tomography, and piezometric pressure*".

We have addressed your concerns either in our detailed responses to the referees or in the revised manuscript, as outlined below:

- U**se of tap water in the lab experiments:** *Please refer to our detailed reply (A6) to Referee #2.*
- **Different instruments used for ERT measurements in 2013/2023**: *We acknowledge the potential uncertainty in dataset comparability and have noted this in the revised manuscript (Line 381).*
- **Selection of ERT dataset from 2013/2023 for inferring seasonal changes:** *Please see our reply (A9) to Referee #1, with corresponding revisions in the revised manuscript (Line 383-399).*
- **Uncertainty in the ERT results due to the inversion procedure:** *We are aware of the potential sources of error in the inversion process. However, our analysis of the raw data indicated a consistent trend with the inverted tomograms, showing decreasing electrical resistivity values from June-September. Therefore, we have decided not to include a detailed analysis of various inversion parameters.*

Please find the updated replies to the reviewers and to the community commenter below. We are looking forward to hearing from you.

With kind regards,

Maike Offer, Samuel Weber, Michael Krautblatter, Ingo Hartmeyer, and Markus Keuschnig
* * *
**Reply to Referee #1**

Dear Referee #1,
Thank you for taking the time to review our submitted manuscript entitled *"Pressurised water flow in fractured permafrost rocks revealed by joint electrical resistivity monitoring and borehole temperature analysis" (egusphere-2024-893).* We greatly appreciate your thorough, constructive and positive feedback, which we have carefully considered. Our detailed point-by-point responses are given below, highlighted in blue, with proposed changes in the revised manuscript indicated in bold. Please note that the lines mentioned in our replies correspond to the revised manuscript. We believe that these revisions and explanations in our responses fully address your concerns and thereby improve the quality and clarity of our manuscript.

Sincerely,
Maike Offer, Samuel Weber, Michael Krautblatter, Ingo Hartmeyer, and Markus Keuschnig
* * *
This reviewer has expertise in permafrost field observations, numerical modelling of frozen soil, and (to a lesser degree) permafrost geophysics.

This manuscript presents a unique dataset of repeated ERT, borehole temperature observations, and site characterization in steep permafrost rock. The combined dataset is beautifully presented and affords insights into the evolution of frozen, thawed, and wet zones in the rock. The careful design of temperature observations allowed detecting fast thermal events at depth that are attributed to water infiltration. These are important topics for research in the context of better understanding permafrost moderated climate control on rock instability.

(A1) Thank you for appreciating the unique, long-term data sets which were achieved under challenging field work conditions as well as the careful design of the results.

The manuscript did not convince me that the data revealed pressurized water as stated in the title. The authors support this inference by mentioning piezometric measurements from late summer 2023 (which are not shown or referenced) and the assumption (which is not developed in detail) that pressurised water flow explains the observed rapid electrical resistivity decline. While I am enthusiastic about the data and many of the analyses presented, a clearer focus, structure, and methodology are required for publication. I recommend encouraging resubmission of this manuscript after adjusting focus and conceptual clarity.

(A2) We value your insightful feedback. Initially, we chose not to include piezometric measurements as the installation was completed in late September 2023, which did not cover the periods of the presented borehole temperature data (01/2016-09/2023) and electrical resistivity measurements (02-06/2013, 09-12/2013, 06-09/2023). In addition, the key seasonal period of snow melt was not covered at the time of our initial submission of the manuscript in March 2023.
However, in response to your suggestion, we included the data set of one piezometer (depth: 16.85 m) from January to June 2024 in the revised manuscript. Therefore, we prepared new subchapters describing the methodology and results. The new designed Figure 8 is included in the revised manuscript, demonstrating an increase in piezometric pressure levels from spring to summer, with maximum heads reaching already up to 11.8 m. These direct observations of water pressure levels strongly support our hypothesis, inferred previously from the electrical resistivity data, that the rock matrix is influenced by pressurized water which is most pronounced in the season of snow melt (i.e. days with average air temperature above 0°C).

[Figure]

**Figure 8.** a) Piezometer pressure (PP) from January to June 2024 recorded near the summit station at a depth of 16.85 m. b) Mean daily air temperature (AT) from the weather station at the Gletscher Plateau (2.940 m asl, distance ~500 m), shown in blue (<0°C) and red (>0°C). The moving mean air temperature (AT) over a 2-hour interval is represented in grey. Yellow bars indicate periods when the mean daily air temperature was above 0°C, during which increases (blue rectangle) and short-term fluctuations with 24-hour frequency (orange rectangle) in pressure level were mostly observed. The grey balk marks a data gap in the weather station recordings.

Regarding your addressed need for a more detailed explanation of how pressurized water flow is revealed from the observed electrical resistivity decline, we elaborated on this in the methodology (*3.3 Laboratory calibration of temperature-resistivity relation*) describing the physical principles of Archie's Law and how we concluded from this to the presence of pressurized water in the discussion section (*5.1 Pressurised water flow in permafrost rockwalls*) to improve the overall conceptual clarity:

**3.3 Laboratory calibration of temperature-resistivity relation**

[revised manuscript text omitted]

Since we present now the piezometric data, we would suggest changing the title of the manuscript to: "***Pressurised water flow in fractured permafrost rocks revealed by electrical resistivity tomography, borehole temperature, and piezometric pressure***".

To achieve a clear structure of the manuscript, the subchapter in methodology and results are restructured to begin with the borehole temperature data, followed by the electrical resistivity observations and laboratory calibrations, and concluding with the piezometric measurements.

- Water flow in fractured permafrost rock has been investigated, and detected with ERT, previously. This study adds to the body of knowledge incrementally. Confident detection of pressurized flow would indeed make it a novel and significant contribution. A more detailed analysis of the thermally detected flow events could likewise be interesting.

(A3) While a more detailed analysis of thermally detected flow events would be indeed interesting, we propose to focus on the novel and significant contribution of the detection of pressurised water flow to maintain the scope and coherence of the manuscript. Including extensive analysis of heat transfer or energy balance would dilute this focus and increase the length of the manuscript, which would be not consistent with your previous and last comment.

- The specific objectives of the research are not clearly articulated. Consequently, the exact state of the art is unclear, the approach and methods cannot be judged in their appropriateness, and the conclusions are not as compellingly underpinned by the evidence presented as they could be.

(A4) To address your concerns, we will introduce the concept of water infiltration in mountain rock slopes in the introduction and explicitly highlight the research gap to clearly articulate the specific objectives of our study:

Line (34-40): "**In some cases, the hydrological processes contributing to the destabilization of the rock slope were directly evidenced by ice and water coating the scarp shortly after the event (e.g., Walter et al., 2020; Mergili et al., 2020; Cathala et al., 2024a).** The infiltration capacity and hydraulic permeability of a rockwall are determined by the degree of fracturing, pore space characteristics, saturation, and temperature (Gruber and Haeberli, 2007). **The surface water availability from snowmelt and rainfall for infiltration into steep rock slopes was recently estimated by a numerical energy and hydrological balance model (Ben-Asher et al., 2023).**"

Line (72-80): "**The influence of water fluxes on the thermal and hydrostatic regime in bedrock permafrost remains a key challenge, despite the demonstrated link between hydrological fluxes and rock slope instability. In this study, we address the lack of quantitative and in-situ observations of rockwall hydrology by analyzing** repeated ERT measurements, ground temperature data from two deep boreholes (2016-2023) and **piezometric pressure data (2024)** at the fractured north face of the Kitzsteinhorn (Hohe Tauern range, Austria). **This study highlights the importance of a higher geoelectrical measurement frequency** compared to annual campaigns, an integrated approach **of different indirect and direct measurements,** and emphasizing the significance of melt period analysis in delineating both temporal patterns and spatial distribution of intense water flow within fractures. **These** insights **will improve our knowledge of the complex seasonal water flow in rockwalls that potentially** accelerate permafrost **degradation and contribute to promoting and triggering factors for rock slope failures.**"

Additionally, we included new piezometric data (as mentioned in A2) and provide a more detailed analysis of the change of the rock temperature regime at depths of 10 and 15 m between 2016-2019 and 2020-22 in borehole B1 (see A9). These additional analyses serve as further indicators, alongside electrical resistivity monitoring, of pressurised water flow and help to compellingly underpin our conclusions.

- Line 300: The cause of the thermal offset stated appears to be speculation. It can equally be explained by transient effects or lateral variation of surface temperature – and neither require invoking thermal effects of water flow.

  (A5) We revise the text to reflect the possible sources of thermal offsets:

  (Line 366-369): *"Beside the possibility of varying subsurface thermal conductivity (Hasler et al., 2011b), this water-flow-induced seasonal succession of rapid warming and slow cooling can result in a positive thermal offset, as observed in B1 and B2 (Fig. 4a), which, over long time periods, results in bottom-up oriented permafrost degradation."*

- Figures 5 and 6: There are strong temporal trends that would provide important background information. Consider showing a depth profile of temporal trends in mean (and maximum?) temperatures over the entire measurement duration. This will help contextualize Figure 4 and its decadal gap.

  (A6) As mentioned in Line 112-113 borehole temperature measurements are available only since December 2015. Therefore, it is not possible to show a temporal trend over the decadal gap between the electrical resistivity observations (2013-2023). Within the available data constrains, we have chosen to focus on the thermal regime in 2023 (Figure 4a, 7) to provide a detailed information of the rock temperature during the electrical resistivity measurements in 2023.

- Line 313: The authors state that obstacles exist for interpreting ERT profiles in fractured rock masses based on laboratory measurements on intact samples (an error). Section 5.1 argues that the differences between lab and field point to pressurized water flow (a signal). Can error and signal be distinguished with sufficient confidence? Explain how.

  (A7) We are aware of the upscaling effect between the electrical resistivity measurements on intact rock sample and on the fractured rockwall at the Kitzsteinhorn, as mentioned in Line 412-416. However, we do not consider the resulting differences between laboratory and field observations as errors. Instead, we interpret them as indicator of conditions at the study site that were not or could not be replicated in the laboratory experiments (i.e. rock masses with water-filled fractures). The full argumentation of how we inferred pressurised water flow from the differences between lab and filed measurements is developed in (A2) and is included in the revised manuscript in section 5.1.

- Line 21: Is permafrost thaw a hazard?

  (A10) We will change it to: (Line 25-26) *"[…] including **rock slope failures from warming permafrost rocks."***

- Section 4.2: What is the impact of using summer ERT that has been measured ten years after the profiles in other seasons? Are we interpreting the influence of seasons or a decade of atmospheric warming (see Figure 6)? This needs to be addressed clearly.

  (A9) As previously mentioned, the novelty and significant contribution of our study lie in the detection of pressurised water flow and seasonal changes. Consequently, we have focused on the results during the thawing season and refrained from a detailed interpretation of decadal permafrost changes. Logistical constraints made it unfeasible to repeat the ERT measurements in 2023, which is why we included the ERT data from 2013. In addition, the ERT measurements from 2023 show that the signal is influenced by the water content of the rockwall, greatly affecting the detection of permafrost and, thus, making it impracticable to draw conclusions about its evolution. We clarify our interpretation by modifying the corresponding sentences in the discussion (section *5.2 Limitations and uncertainties*) and include an explanation of the differences in the results from June/September in 2013 and 2023:

  (Line 385-399): *"**Atmospheric conditions vary slightly between years, affecting the timing of snow melt and hence the change in resistivity regime. The ERT results from June 2013 exhibit higher resistivity values in the upper part of the profile (> 64 kΩm) compared to the ERT measurement in 2023 (Fig. 7), probably due to colder rock and atmospheric conditions prior to the measurement (Fig. A1).** The penetration depth of the current flow into the subsurface depends on the characteristics of the top layer and may vary seasonally. Dry and frozen conditions can impede current flow, while water-saturated conditions might trap current flow, resulting in an attenuated current flow into deeper layers (Loke, 2022). Poor electrode coupling is often associated with frozen conditions and ice-filled fractures and cracks from autumn to spring, which cause noisy data and can lead to inversion artifacts represented by a high RMS error. **This phenomenon was observed, for example, in September 2013 vs 2023, where cold air temperatures likely caused freezing of the rock surface layer prior to the ERT measurement (Fig. A1), resulting in high contact resistance and an inability to resolve the long-lasting summer thermal signal in greater depths.** Consequently, we refrained from detailed interpretation of the inverted tomograms from February to May 2013 and September to December **2013 but included the ERT data to cover all season. We assume that a repeat of the ERT measurements in 2023 during the frost period would have yielded comparable results, as borehole temperature show frozen subsurface conditions from January to June and from October to December 2023 (Fig. 7), during which no thermal anomalies or irregularities were observed (Fig. 5).**"*

- Section 5.3: Some of the statements seem rather confident. They could be shortened and made specific to well supported conclusion and, as such, added as a short outlook paragraph to the conclusion. Some of the other text in the section is better suited for the introduction of a paper.

  (A10) We decided to discuss our findings in detail in section 5.3, as we believe they have significant implications for rock wall instabilities and, more generally, for high alpine permafrost monitoring routines. We have already focused on specific processes and ensured that our statements are well-supported by literature and/ or our study. However, we followed your suggestions and integrated some of the statements from section 5.3 into the conclusion:

Line (446): " [...]
4. *Monitoring of alpine permafrost often relies solely on annually repeated geoelectric measurements, mainly due to complicated logistics and harsh measurement conditions. However, our study suggests that higher ERT measurement intervals are required to decipher the complexity of hydrothermal processes in permafrost rockwalls and fully assess the rate and extent of permafrost evolution. Monthly repeated measurements in this contribution represent a significant advancement compared to annual surveys.*
5. *We emphasize the key role of complementary temperature measurements and their joint analysis. Low electrical resistivity values in the absence of borehole temperatures may be misinterpreted as permafrost-free rock slopes, yet. They could serve as an indicator of water-saturated conditions above a potential permafrost body.*

*This study has broad implications for understanding hydrothermal processes in steep, fractured rock walls, **profoundly impacting** the rate and extent of permafrost degradation and related hazards. **Future developments are needed to validate and quantify our observations. Of particular interest would be simultaneous electrical resistivity and piezometric measurements during the thawing season, whereby daily or hourly observation intervals would represent another significant step towards a better understanding of the transient nature of water flow in fractures.***"

- The manuscript text should be shortened and edited for clarity in structure and arguments. Some of the referencing could be tightened, giving preference to one good reference backing up a particular argument instead of listing a handful of publications.

  (A11) As mentioned in (A2), we reordered the subchapters in the methodology and results to improve the clarity and structure. These revisions, along with the other modifications, significantly strengthen our arguments, particularly through the newly designed Figures 8 and 6 (see author comments for referee 2) and the newly included results from piezometric measurements.
* * *
**Reply to Referee #2**

Dear Referee #2,
Thank you for your review of our manuscript *"Pressurised water flow in fractured permafrost rocks revealed by joint electrical resistivity monitoring and borehole temperature analysis" (egusphere-2024-893).* We sincerely appreciate your comprehensive, constructive and positive feedback. We have carefully considered your comments, and our detailed responses are provided below, highlighted in blue, with proposed changes in the revised manuscript indicated in bold. Please note that the lines mentioned in our replies correspond to the revised manuscript. We believe that these revisions and accompanying explanations effectively address your concerns and thereby improve the quality and clarity of our manuscript.

With kind regards,
Maike Offer, Samuel Weber, Michael Krautblatter, Ingo Hartmeyer, and Markus Keuschnig
* * *
The paper entitled "*Pressurised water flow in fractured permafrost rocks revealed by joint electrical resistivity monitoring and borehole temperature analysis*" presents a combination of repeated electrical resistivity tomography (ERT) data and borehole temperature data on a high mountain rock wall site in Austria to discuss potential effect of pressurised water on rock wall temperature and destabilization.

The paper addresses an important topic in high alpine permafrost and geomorphology community that is the characterization of water flows and their impacts for permafrost dynamics and morphodynamics. Addressing this topic is challenging due to the difficulty to observe and measure water flow processes in high mountain and the non-linearity of the related processes. Geophysical approaches are for sure one of the most promising method to investigate these processes.

Overall, the paper is well written and well structured. The ERT dataset is also quite unique and was gained through challenging field work. The figures are very nice and clear. However, I find some major limitations and I would recommend publication after major revisions.

(A1) Thank you for appreciating the uniqueness of the dataset and the careful design of the figures.

GENERAL COMMENTS

One of the major issue is that some of the main findings that are reported are not appropriately demonstrated (see further comments). Furthermore, in the current state, I find it difficult to understand what is the novelty of the conclusions of this paper that echoes a former paper from Keushing et al. (2017). Therefore, I am not convinced by the last sentence of the abstract, especially by the expression "shows for the first time".

(A2) In the revised manuscript, we addressed your concerns by including the missing main findings, specifically piezometric measurements (see detailed response (A13)). While Keuschnig et al. (2017) suggested that pressurised water flow warms the surrounding rock

upwards, their conclusions were based primarily on fracture inventories, visual observations of cleftwater and near-surface temperature measurements. The focus of the former paper was on testing automated ERT as an early warning system, and it did not include geophysical observations during the peak season of water flow (i.e. snowmelt season between June and September) nor deep borehole temperature and piezometric measurements. Our study, however, incorporates these critical observations, which are necessary to substantiate the hypothesis of water flow and its related processes. In the submitted manuscript we "show for the first time" a comprehensive analysis of direct and indirect observation methods which enables a characterization of the thermal and spatial impact of water flow, and re­garding slope stability the build-up of critical hydrostatic levels.  In response to your con­cerns, we revised the abstract to highlight the novel contribution of our study more precisely and to distinguish our work from the earlier study by Keuschnig et al. (2017). We modified the last sentence to:

*(Line 19-23)* **"This study provides for the first time direct and indirect observations of pres­surised water flow which shows that, in addition to slow thermal heat conduction, perma­frost rocks are subjected to sudden push-like warming events and long-lasting rock tem­perature regime changes, favouring accelerated bottom-up permafrost degradation, and contributing to the build-up of hydrostatic pressures potentially critical for rock slope sta­bility."**

A detailed explanation for the mentioned "long-lasting rock temperature regime changes" is given in (A13).

INTRODUCTION/ STUDY SITE

Since the core of the paper is about water infiltration I would suggest to better introduce water infiltration in rock slopes (see Hasler et al., 2011a, Ben-Asher et al., 2023) and in mountain permafrost ground in general.

(A3) We agree that a more thorough introduction of water infiltration in rock slopes will en­hance the focus and structure of the manuscript. We already mentioned Hasler et al., 2011a in our introduction, but we incorporated the other suggested literature (Ben-Asher et al., 2023, Cathala et al., 2024) and include a concise paragraph on this topic in the introduction:

*Line (34-40):* *"***In some cases, the hydrological processes contributing to the destabilization of the rock slope were directly evidenced by ice and water coating the scarp shortly after the event (e.g., Walter et al., 2020; Mergili et al., 2020; Cathala et al., 2024a).*** The infiltra­tion capacity and hydraulic permeability of a rockwall are determined by the degree of frac­turing, pore space characteristics, saturation, and temperature (Gruber and Haeberli, 2007). ***The surface water availability from snowmelt and rainfall for infiltration into steep rock slopes was recently estimated by a numerical energy and hydrological balance model (Ben-Asher et al., 2023).***"*

A climate and weather analysis during the measurement period would be highly welcome in the site description, especially for discussing the results afterwards (see further comments). An option would also be to make a general section about Study site and instrumentation as I

missed some information about the boreholes (depth, available time series…) and the ERT system (length, number of electrods…).

(A4) To maintain the clear focus of the manuscript, we acknowledge the benefits of providing a weather analysis during the measurement period (2013, 2023) and suggest incorporating this without extending a full climate analysis. We will include the air temperature, precipitation, and snow height recordings from the weather station "Gletscherplateau" (2.940 m asl, distance ca. 500 m) and "Alpincenter" (2.450 m asl, distance < 2 km) in a new Figure A1. However, the weather analysis is provided in the Appendix, since we already show the mean daily air temperature, the estimated snow cover area and snow height at the ERT measurement days in 2023 in the main text in Figure 2. Since the focus of the manuscript is on the seasonal water flow during the thawing period (June-September) and not on decadal permafrost changes (see also A12), the atmospheric conditions during the ERT observations in 2023 are of particular interest. Additionally, we will present air temperatures from January to June 2024 in Figure 8 to correlate with the observed water pressure levels.

[Figure]

**Figure A1.** Mean monthly air temperature (MMAT) and the weekly moving average for the air temperature for 2013 and 2023 were recorded at the nearby weather station at the Glacier Plateau at 2.940 m asl (~ 500 m distance from the study site). The MMAT in 2013 and 2023 showed comparable values for most months (ΔMAAT < 2.5 °C), with notable differences in February (2013: -13.1°C/2023: -7.9 °C), April (2013: -3.2 °C/2023: -6.2 °C), and September (2013: 1.8 °C/2023: 6.5 °C). Snow height was also recorded at the Glacier Plateau weather station. Although the slope angle at the weather station is lower than the surrounding the study site, the orientation is the same, making the snow height trends and subsequent snowmelt infiltration patterns transferable to the study site, albeit not the absolute values. Snow height in 2013 and 2023 is visualized with a solid line, with interruptions due to data gaps. The snowmelt was most pronounced in late May 2013 and in June 2023. The bar plot shows the daily rainfall sum at the weather station at the Alpincenter (2.450 m asl, distance < 2 km). Notably, no heavy rainfall events were registered shortly before or on the days of ERT measurements.

We have decided to provide detailed information about the ERT system and boreholes within the methodology, specifically in sections *3.1 Borehole temperature measurements and 3.2 ERT data acquisition:*

ERT system information: Length/ number of electrodes:

- (Line 153): *"[…] 37 electrodes were permanently drilled into the bedrock at intervals of 2 m."*
- (Line 162-163): *"[…] only the top 30 fixed electrodes could be used in 2023 **(i.e. 58 m profile length)**."*

Borehole information: Depth/ time series:

- (Line 164-165): *"Both boreholes were drilled perpendicular to the surface and schistisity, reaching a depth of 22 m (B1) and 30 m (B2)."*
- (Line 124-125): *"Sensors were installed at 0.1, 0.5, 1, 2, 3, 5, 7, 10, 15, 20, 21.5 (only B1), 25 (only B2) and 30 m (only B2) depth."*
- (Line 112-113): *"Rock temperature measurements in two deep boreholes […] have been conducted since December 2015 **and were analyzed until December 2023.**"*
- (Line 128-132): *"Lightning strikes damaged several thermistors throughout the long-term monitoring (2016-2023), leading to data gaps starting in June 2017 (B2 = 0.5, 10 m), July 2019 (B2 = 20 m), June 2020 (B2 = 7 m), September 2020 (B2 = 25 m), April 2023 (B1 = 20 m), June 2023 (B1 = 7, 10, 21.5 m, B2 = 1 m). Warming releases resulting from construction activities in summer 2023, mainly due to drilling operations near B1, could have affected ground temperature measurements. Consequently, we excluded the affected data sets from B1 (August-December 2023)."*

I also wondered why the ERT data and temperature data are not directly compared and discussed since the resistivity values could be used to infer temperature values based on the lab results.

(A5) Giving the constraints of the borehole data, which began in December 2015, a direct comparison of ERT data and borehole temperature is only feasible for the observations in 2023. We have addressed this comparison in Figure 7 by indicating the depth of 0 °C in the tomograms, and in greater detail in Figure 9. In Figure 9a, we focus on the 0 °C and -0.5 °C isotherms (i.e., permafrost table) observed in borehole B2 and compare these with the median resistivity values of the ERT measurements conducted from June to September 2023. Figure 9b illustrates the discrepancy between the temperature-resistivity relationship derived from the lab and field measurements. We thoroughly discuss these in section 5.1, which also forms an important basis of our argument for a widespread water flow injection into the fracture network (see also (A13)).

METHODS

Has tap water some implications on the freezing point? Duvillard et al. (2021) showed that it has a different freezing temperature than snowmelt water that is more representative of the natural environment. I would have first presented the field before the laboratory calibration approach as the latter completes the former.

(A6) Thanks for your remarks regarding the laboratory experiments. We are aware that tap water can have minor implications on the freezing point. Duvillard et al. (2021) conducted electrical conductivity experiments only on one sample saturated with tap water and on one

with snow melt, making it difficult to quantitatively separate the bias due to rock heterogeneity (i.e., number of interconnected pores) and water type. In our laboratory study, we tested seven different rock samples with the consistent preparation and measurement procedures, demonstrating the influence of rock heterogeneity by revealing a range of freezing points between 0.22 °C and -0.31 °C (see Table D1). Regarding your concerns, we determine the conductivity (i.e., concentration of ions) of the snow melt water from the field site to 0.014 S m$^{-1}$, which is in the same order of magnitude from the used tap water with 0.058 ± 0.002 S m$^{-1}$ (Line 193). Comparing the lab experiments with the field observation from June 2023 (Figure 9b), we observed a slight shift of the freezing point to approximately -1.0 °C, likely influenced more by the fractured rock mass rather than snow melt water. This uncertainty is address with the defined transition zone in the ERT color scheme, and we refrain from directly linking electrical field resistivity values to temperature.

To strengthen the focus of the paper, we followed your suggestion and restructured the subchapters of the methodology and results, beginning with borehole temperature observations, followed by the field ERT measurements and concluding with laboratory calibrations.

The ground contact resistance is not presented while it is a major parameter of the measurement as explained by Herring et al. (2023). But some datasets have huge RMS and this could be partly due to poor contact resistance. This part of the work has to be described and addressed.

(A7) We agree that high contact resistance (i.e., poor contact between electrodes and bedrock) limits the current injection into the bedrock and degrade data quality. As noted by Herring et al. (2023), contact resistances vary with site conditions and the season. In frozen bedrock, high contact resistances remain a key challenge, which can be mitigated e.g. by adding fresh water. However, given that the investigated rockwall is snow-covered during the winter months, it is not feasible from a logistical and safety perspective to access the corresponding electrodes. Our manuscript primarily focusses on the ERT measurements conducted in summer 2023, when contact resistance were low (mostly < 200 kΩ, which are accepted values according to Herring et al. (2023)) and RMS values were minimal (Table B1: 3.6-4.3 %). In the revised manuscript, we included the contact resistivity values of the respective measurements in Table B1. Although the contact resistances were higher during the observations from the winter months, resulting in higher RMS values, we decided to include these ERT measurements to provide a comprehensive dataset covering all seasons.

We addressed this issue in the revised manuscript in section 3.2 (Line 178-186):

Line (178-186): *"Resistivity models with a high root mean square (RMS) error between the modelled and observed data are obtained, particularly during the winter months when frozen surface conditions impede the coupling of electrodes.* **Contact resistance values vary seasonally (Herring et al., 2023); in our case, values < 200 kΩ were observed during summer measurements, while values > 200 kΩ were mainly measured during winter measurements (Table B1). The high contact resistances (> 200 kΩ) could not be mitigated due to safety issues of accessing the problematic electrodes during the snow-covered period of the rock face.** *However, an assumption of inaccuracy and subsequent complete exclusion of the affected data sets***, justified** *by high contact resistances and high RMS errors***, would make it impossible to provide all-season ERT observations.** *We, therefore, retained the data sets of*

*the winter measurements **(February-May and September-December 2013)** but withheld detailed interpretation in recognition of the potential presence of noise. "*

From L175, images of the rock discontinuities are mentioned but not displayed in any way. That is a pity because the paper attempts to link ERT data to rock discontinuity data. That would be interesting to better show these data.

(A8) Since our discussion relies on the linkage between ERT, piezometric data (will be included in the revised manuscript, see (A13)) and rock discontinuities, we have carefully described the geotechnical setting of the rock face and presented the rock discontinuities data in several sections:

- Section 2 (Study site): Figure 1 shows the geotechnical setting of the rock face by a schematic representation of the main discontinuities and the dip angel and direction of their mean set planes. Additionally, we described the characteristics of the joint sets in the text (Line 87-91):
  *"The north face of the Kitzsteinhorn exhibits a significant degree of fracturing, characterized by joint openings of up to 20 cm, predominantly along cleavage planes (Schober et al., 2012). The development of the enormous number of joint sets was favored by stress release and intense physical weathering processes (Hartmeyer et al., 2012). The main joint sets are K1, which has a sub-vertical dip to the west, and K2, which features a steep dip to the southwest (Fig. 1). K3 and K4 are less abundant. K3 dips medium-steeply to flat to S-SSE, and K4 dips steeply to NW."*
- Section 4.1 (Borehole temperature and thermal anomalies): As mentioned in Line 125-126 (section 3.1), optical borehole scanning was performed to identify and locate discontinuities. Instead of simply showing the images, we analyzed the scans and presented the results in Figure 5, illustrating the schistosity and cleft locations along the borehole B1 and B2.  Here we linked the occurrences of discontinuities with the frequency of thermal anomalies (Line 261-265): *"Optical borehole imaging shows a pronounced occurrence of clefts with apertures of up to 5 cm in the first ten meters in B1 and B2 and intact rock mass of calcareous mica-schist with marked schistosity at greater depths (Fig. 5). Thermistors installed close or within clefts or areas of schistosity exhibit a higher frequency of thermal anomalies, as evidenced by the counts recorded (e.g., B1-2 m: 18, B2-3 m: 25), in contrast to thermistors installed at a greater distance, not exceeding 50 cm, from discontinuities (B2-2 m: 10, B1-3 m: 16)"*
- Section 5.1 (Pressurised water flow in permafrost rockwalls): Figure 10 shows an image of one prominent K2 joint with apertures around 5 cm, which may play a particular role for the infiltration of water, as described from Line 342-348.

The calibration is based on an intact rock sample while the paper focuses on specific processes of fractured rock. The fractures might not be entirely filled with water or ice and this is not discussed. The signal of air and ice is the same, and this needs to be discussed and clarified. This means that the results must be considered with caution as well.

(A9) Thanks for your comment, we recognize the importance of distinguish between air and ice signals, as both can produce signals in the same range. To address this, we clarify this point in the results and discussion section, chapter 4.2 and 5.1:

-   Line 286-287: *"Nevertheless, electrical resistivity values remained above 32 kΩm, in-dicating frozen rock and **air-/**ice-filled joints."*
-   Line 320-321: *"High-resistivity sections above 19 kΩm indicate frozen conditions with **air-/**ice-filled joints [...]"*
-   Line 362-363: *"[...] we suggest that fractures can act as cooling pathways, favoring the formation of freezing corridors **and air ventilation."***

Despite this, we assume high-water levels in the fractures from July to September since the drastic decline of the electrical resistivity values of more than an order of magnitude can only be explained by a widespread water infiltration into the fracture network (see also A13).

Rather than number of electrods, I would find it more convenient to speak in terms of distance along the profile (see also comment on the lack of information on the profile length).

(A10) We modified the corresponding passages and referred to the distance along the profile rather than number of electrodes.

The calculation of the thermal anomalies must be clearly detailed in the Methods section as this is a central part of the investigation.

(A11) In response to your suggestion, we provided a description of the determination of thermal anomalies in the methods section (3.1) and shorten the respective sentences in the results section (4.1):

*Line (133-142):* ***"The thermistor signals in B1 and B2 were analyzed for irregularities and characteristics typical for non-conductive heat transfer. Near-surface temperatures (depth < 2 m) were excluded from the analysis as they are characterized by short-term fluctuations with large amplitudes, making distinguishing between changes induced by non-conductive heat transfer and meteorologically forced changes complex. For the recordings in 2 and 3 m depth, thermal anomalies were identified using the first derivative with a moving average of 12 points (i.e., measurement interval of 2 hours). High signals in the first derivative were manually reviewed for characteristics typical of non-conductive heat transfer, which exhibit a temperature rise of up to 0.7 °C in less than 2 hours. Sudden, significant changes between two measurements (10 min) with a return to the previous temperature level are caused by overvoltage effects following lightning strikes and were therefore not considered further. Due to the smooth curvature of the thermal signals in 5 m depth, thermal anomalies were directly visible in the data and were manually determined."***

RESULTS

Looking at Figure 4, I wonder how the results from Sep/June 2013 and Sep/June 2023 can be so different? Why don't we see the summer signal reaching 10 m depth in early winter? Could the top part of the profile with relatively high resistivity values during the thawing season could be attributed to desiccation (see also comment on air signal)? The decadal permafrost change could be detailed and discussed to take full advantage of the presented data.

(A 12) We included a short paragraph in the discussion section 5.2 to explain the difference between the results from June/September in 2013 and 2023:

(Line 385-399): *"Atmospheric conditions vary slightly between years, affecting the timing of snow melt and hence the change in resistivity regime. The ERT results from June 2013 exhibit higher resistivity values in the upper part of the profile (> 64 kΩm) compared to the ERT measurement in 2023 (Fig. 7), probably due to colder rock and atmospheric conditions prior to the measurement (Fig. A1). The penetration depth of the current flow into the subsurface depends on the characteristics of the top layer and may vary seasonally. Dry and frozen conditions can impede current flow, while water-saturated conditions might trap current flow, resulting in an attenuated current flow into deeper layers (Loke, 2022). Poor electrode coupling is often associated with frozen conditions and ice-filled fractures and cracks from autumn to spring, which cause noisy data and can lead to inversion artifacts represented by a high RMS error. This phenomenon was observed, for example, in September 2013 vs 2023, where cold air temperatures likely caused freezing of the rock surface layer prior to the ERT measurement (Fig. A1), resulting in high contact resistance and an inability to resolve the long-lasting summer thermal signal in greater depths."*

From Line 293-295, we pointed out our interpretation of the relatively high resistivity at the top part of the profile during the thawing season (July to September):

Line (293-295): *"The upper part of the profile (∼ x=0-18 m) is unaffected by the trend. A high resistivity body of ≥ 32 kΩm remains stable during the summer month, probably due to the shielding of water infiltration through the cable car station and consequent desiccation of the surface rock layer, combined with an intact rock mass without major fractures."*

Since the novelty of our manuscript is to reveal pressurised water flow during the thawing period, we refrained from a detail interpretation of the decadal permafrost change. Our initial motivation to include the data from 2013 was to cover all seasons, including the frost period, which was logistically not feasible in 2023, but would probably have yielded comparable results. We included a short paragraph in the discussion section 5.2 to clarify our choice of ERT data and suggest to refraining from a detailed interpretation of the decadal permafrost change, since we did not present ERT measurements from the same seasonal periods nor a decadal recording of borehole temperature or piezometric data (Figure 8). In addition, the ERT measurements from 2023 show that the signal is influenced by the water content of the rockwall, greatly affecting the detection of permafrost and, thus, making it impracticable to draw conclusions about its evolution.

(Line 395-399): *"[…] Consequently, we refrained from detailed interpretation of the inverted tomograms from February to May 2013 and September to December 2013 but included the ERT data to cover all season. We assume that a repeat of the ERT measurements in 2023 during the frost period would have yielded comparable results, as borehole temperature show frozen subsurface conditions from January to June and from October to December (Fig. 7), during which no thermal anomalies or irregularities were observed (Fig. 5)."*

DISCUSSION

The contradiction with the Archie law is weak as the law is not presented nor discussed in the paper. The same is true with the piezometer data. That is a pity to mention such data without using them extensively nor showing them. L286-288: I do not fully agree with the statement "high impact … on thermal processes". The study shows only short term and minor temperature changes, but great changes in the electrical resistivity that is by essence strongly sensitive to water changes. I would suggest using more balanced wording or to strengthen the demonstration.

(A13) As the missing piezometer data was also addressed by Referee #1, we here include our response to Referee #1 which also concern the contradiction with the Archie Law:

*Initially, we chose not to include piezometric measurements as the installation was completed in late September 2023, which did not cover the periods of the presented borehole temperature data (01/2016-09/2023) and electrical resistivity measurements (02-06/2013, 09-12/2013, 06-09/2023). In addition, the key seasonal period of snow melt was not covered at the time of our initial submission of the manuscript in March 2023.*
*However, in response to your suggestion, we included the data set of one piezometer (depth: 16.85 m) from January to June 2024 in the revised manuscript. Therefore, we prepared new subchapters describing the methodology and results. The newly designed Figure 8 is included in the revised manuscript, demonstrating an increase in piezometric pressure levels from spring to summer, with maximum heads reaching already up to 11.8 m. These direct observations of water pressure levels strongly support our hypothesis, inferred previously from the electrical resistivity data, that the rock matrix is influenced by pressurized water which is most pronounced in the season of snow melt (i.e. days with average air temperature above 0°C).*

[Figure]

**Figure 8.** *a) Piezometer pressure (PP) from January to June 2024 recorded near the summit station at a depth of 16.85 m. b) Mean daily air temperature (AT) from the weather station at the Gletscher Plateau (2.940 m asl, distance ~500 m), shown in blue (< 0°C) and red (> 0°C). The moving mean air temperature (AT) over a 2-hour interval is represented in grey. Yellow bars indicate periods when the mean daily air temperature was above 0 °C, during which increases (blue rectangle) and short-term fluctuations with 24-hour frequency (orange rectangle) in pressure level were mostly observed. The grey balk marks a data gap in the weather station recordings.*

*Regarding your addressed need for a more detailed explanation of how pressurized water flow is revealed from the observed electrical resistivity decline, we elaborated on this in the*

*methodology (3.3 Laboratory calibration of temperature-resistivity relation) describing the physical principles of Archie's Law and how we concluded from this to the presence of pressurised water in the discussion section (5.1 Pressurised water flow in permafrost rockwalls) to improve the overall conceptual clarity:*

**3.3 Laboratory calibration of temperature-resistivity relation**

**(Line 211): " [...] The electric properties of water-saturated rocks is determined by the ionic transport in the liquid phase and, therefore, by the amount of interconnected pores. The well-known empirical law develop by Archie (Archie, 1942) relates the resistivity ρ to the functional porosity φ, the resistivity of the pore water $\rho_w$, and the fraction of the pore space occupied by liquid water S:**

$$\rho = a\varphi^{-m}S^{-n}\rho w$$

**where a, n and m are empirically determined constants. At subzero temperatures and under partially frozen conditions, the electrical properties of the rock depends on the remaining unfrozen water content in the pores. As the temperature drops to the equilibrium freezing temperature, pore water saturation decreases while the resistivity of the pore water also decreases due to the migration of electrolytes from the freezing water to the remaining unfrozen water content, resulting in increased electrolyte concentration. Above the equilibrium freezing temperature, the resistivity of the rock is indirectly related to temperature changes, as temperature affects the mobility of the solute electrolytes."**

*5.1 Pressurised water flow in permafrost rockwalls*

*The unique time series of laboratory-calibrated ERT observations presented in this paper enable a quantitative interpretation of seasonal changes in frozen rockwalls. High-resistivity sections above 19 kΩm indicate frozen conditions with ice-filled joints during the frost season (October-May). Slight warming of the rock surface after the snow cover disappears in late spring is indicated by decreasing resistivity at shallow depth (e.g., tomography from June 2023 in Fig. 7). Ice-filled joints probably act as an aquitard, constraining deep infiltration into the joint system, with snowmelt mainly draining along the rock surface. From June to July, rapid changes in resistivity of more than one order of magnitude were observed at ~1–7 m depth coincident with a borehole temperature warming accompanied by active layer deepening from 1.7 to 2.7 m depth between the ERT measurement dates in June and July (Fig. 9a). The low resistivity zone (~4 kΩm) in July in the lower part of the tomography (~x = 28-58 m) gradually expands to higher rock slope sections (Fig. 4) and to the bottom of the ERT profile until September (~10 m depth, Fig. 9a), while the 0 °C/–0.5 °C isotherm (i.e., permafrost table) changes marginally (Aug: 3.5/4.1 m, Sep: 3.5/4.3 m, **Fig. 9a**).*
***The term 'pressurised' here refers to a piezometric head of a few meters. The rapid resistivity decline observed suggests pressurised nature of water flow in fractures, supported by additional evidence. This evidence comprises visually observed water outflow from fractures (Fig. 10) and first piezometric measurements showing rapidly increasing pressure levels in the thawing season, with piezometric heads reaching up to 11.8 m (Fig. 8). Without assuming pressurised flow, the decline in electrical resistivity from July to September (Fig. 7,9) would be inconsistent with Archie's law. In thawed conditions, resistivity decreases for various rock types at a rate of ~2.9 ± 0.3 %/°C (Krautblatter, 2009), and ac-***

*cording to our laboratory calibrations, by 4.5 ± 0.3 %/°C (Fig. 3, Table D1). Thus, a temperature warming from July to September (Fig. 9) in already fully saturated rock with constant porosity would not cause a significant further and rapid electrical resistivity decline. This can only occur if pressurised water flow contributes to additional hydraulic opening of fractures within days to weeks. In addition, the coincident rapid changes (Fig. 5) and regime changes in rock temperature (Fig. 6) cannot be explained solely by diffusive heat exchange (Noetzli et al., 2007; Krautblatter et al., 2010), but only by water flow in open fractures (Phillips et al., 2016), facilitating a thermal shortcut between the atmosphere and the subsurface (Hasler et al., 2011a).*
*We hypothesize that […]"*

Since we present now the piezometric data, we would suggest changing the title of the manuscript to: "***Pressurised water flow in fractured permafrost rocks revealed by electrical resistivity tomography, borehole temperature, and piezometric pressure***".

To achieve a clear structure of the manuscript, the subchapter in methodology and results is restructured to begin with the borehole temperature data, followed by the electrical resistivity observations and laboratory calibrations, and concluding with the piezometric measurements.

Figure 6 demonstrates the *"high impact of fluid flow in fractures on […] thermal processes"* by showing the thermal signal of borehole B1 in 10 and 15 m depth. We acknowledge that the current layout might the long-term temperature warming effects are rather underrepresented and likely to be overseen. Therefore, we have designed a new Figure 6, which is included in the revised manuscript. This figure aims to clarify the need to better understand these observed temperature regime changes and to strengthen our conclusions about the major implications of fluid flow on the thermal regime.

[Figure]

*Figure 6. Borehole temperature at depths of 10 and 15 m between 2016 and 2022: a) Thermal signals in 10 m depth, with minimal values highlighted (top) and at 15 m depth, with maximal values highlighted (bottom). b) Mean monthly temperature values, with each ring representing a measurement year and the radius increasing for more recent observation years.*

Another point that comes to my mind is the effect of anisotropy in such type of rock with a high degree of schistosity. This could be at least discussed and ideally investigated through lab measurements.

(A14) We considered the effect of anisotropy in the section *5.2 Limitations and uncertainties*:

(Line 412-416): *"However, the problem of extrapolating from laboratory experiments to field observations (Zisser et al., 2007; Krautblatter et al., 2010) was highlighted by our ERT observation of highly fractured rock face **with anisotropic characteristics**, which indicated the strong influence of water-saturated fractures and cracks on the electrical properties, less represented by the intact rock mass of the laboratory studies."*

CONCLUSION

The first point rather reminds the initial hypothesis than bringing a demonstration of its validation. In my opinion, the 3rd and 4th points are not demonstrated in the paper.

(A15) We modified the listing in the conclusion by considering your concerns, the suggestions of Referee 1, and the new included piezometric data set:

*(Line 436): "[...] by combining repeated electrical resistivity monitoring, long-term temperature measurements in deep boreholes, **and piezometer observations.** The following conclusions are drawn:*

1. *A massive decrease in electrical resistivity values during the thawing season (July-September) can be indicative for snow melt water infiltration into the rockwall draining along the schistosity and interconnected joints, and subsequently becoming pressurised within a widespread fracture network.*
2. *Hydrostatic pressure levels of up to 11.8 m indicate a widespread water infiltration into the fracture network, which potentially alters slope stability by favouring bottom-up permafrost degradation.*
3. *Small, abrupt temperatures anomalies registered in the two boreholes (2.0, 3.0 and 5.0 m depth) suggest non-conductive heat flux in fractures. Frozen rock is warmed more rapidly by these sudden push-like events of heat transport from the surface than by slow thermal conduction alone.*
4. *Long-term regime temperature changes were identified in two boreholes in 10 and 15 m depth between 2016-2019 and 2020-2022, indicating the pronounced heat transfer by infiltrating water.*
5. *Monitoring of alpine permafrost often relies solely on annually repeated geoelectric measurements, mainly due to complicated logistics and harsh measurement conditions. However, our study suggests that higher ERT measurement intervals are required to decipher the complexity of hydrothermal processes in permafrost rockwalls and fully assess the rate and extent of permafrost evolution. Monthly repeated measurements in this contribution represent a significant advancement compared to annual surveys.*
6. *We emphasize the key role of complementary temperature measurements and their joint analysis. Low electrical resistivity values in the absence of borehole temperatures may be misinterpreted as permafrost-free rock slopes, yet. They could serve as an indicator of water-saturated conditions above a potential permafrost body.*

*This study has broad implications for understanding hydrothermal processes in steep, fractured rock walls, **profoundly impacting** the rate and extent of permafrost degradation and related hazards. **Future developments are needed to validate and quantify our observations. Of particular interest would be simultaneous electrical resistivity and piezometric measurements during the thawing season, whereby daily or hourly observation intervals would represent another significant step towards a better understanding of the transient nature of water flow in fractures.**"*

Detailed comments

- Abstract L1: failures do not occur from permafrost itself as permafrost is by definition a temperature, rather use permafrost ground or permafrost-affected slope.
  We modified it to *"**permafrost rocks".***

- L 140: what "representative" means here?"
  We will rephrase the sentence without "representative".

- L 148: do you mean average values? (positive values for all days)

We will change it to *"[…] indicated positive mean values […]"*

- L 195: here consider the comment about air and ice signal
  See comment (A9). → *"[…] air-/ice-filled joints."*

- L218: where do we see the mentioned zero-curtain? This is crucial to see it and how long it lasts as it provides an information about the ice content.
- The zero-curtain effect can be seen in several thermistor signals in Figure 5. → Line 280-281: *"The zero-curtain effect […] was most pronounced at 3 m depth in B2 (Fig. 5)."*

- L 223: which construction activity are you talking about?
  We provided more details → *"[…] the year before construction activities **for summit station maintenance close to the borehole** (August 2022-July 2023) was analysed."*

- L225: "thermal offset" is not an appropriate concept for rockwalls, see Hasler et al., 2011b
  L227-228: the explanation of the "thermal offsets" is not clear
  We recognize that the concept of the "thermal offset" is considered impractical for fractured bedrock due to the high variable mean annual ground surface temperature and active layer thickness, as noted by Hasler et al. (2011b). However, in our specific case, boreholes B1 and B2 both observed similar active layer thickness (B1=4.3 m, B2=3.9 m, see Line 246), suggesting relatively stable depth of the permafrost table in the investigated rockwall. Therefore, we propose using the concept of "thermal offset" in this context, while acknowledging and clearly indicating the limitations in our interpretation:

  (Line 247-253): ***"The thermal offset, defined by the difference between the annual mean ground surface temperature (i.e., temperature recording in 0.1 m depth) and the temperature at the permafrost table (Burn and Smith, 1988), is generally considered impracticable for fractured bedrock due to the high variable microclimate and active layer thickness (Hasler et al., 2011b). However, in our specific case, both boreholes suggest that the permafrost table depth are within a similar range. Therefore, considering the potential variability in microclimate, we propose that the concept of thermal offset is practicable and demonstrate positive values (B1 = 1.5 °C, B2 = 0.9 °C)."***

  (Line 366-369): ***"Beside the possibility of varying subsurface thermal conductivity (Hasler et al., 2011b),*** *this water-flow-induced seasonal succession of rapid warming and slow cooling can result in a positive thermal offset, as observed in B1 and B2 (Fig. 4a), which, over long time periods, results in bottom-up oriented permafrost degradation."*

- L230: calculation of these abrupt changes must be clearly explained in the method section
  See comment (A11).

- L233: how is this threshold of values defined?
  See comment (A11): The first derivatives were manually reviewed for high signals and marked in Figure 5, which yielded a temperature rise of up to 0.7 °C in less than 2 hours.

- L296-297: and what about air?
  See comment (A9). We modify it to → *"[…] we suggest that fractures can act as cooling pathways, favoring the formation of freezing corridors **and air ventilation.**"*

Beyond this, the major issue that I have with this manuscript lies in the ERT dataset selection. Due to some lightning strikes, most ERT measurements between June and September 2013 were corrupted and the authors decided to fill the gap with data from 2023, 10 years after the original measurements. The authors are comparing monthly ERT measurements coming from two sets of measurements spaced by 10 years, and do not address the issues created by such a significant temporal gap. As they correctly put it in their introduction, the rise of temperatures and the permafrost degradation have accelerated in the last decade, and there is little reason to believe that the study site has not been affected too. In fact, it is

clear from Figure 4 that the resistivity of the bedrock along the survey line has changed tremendously between 2013 and 2023 during the months of June and September (the only months measured both in 2013 and 2023). Visually, the most impressive difference comes from September where the resistivity of the whole cross section is about 2 orders of magnitude smaller in 2023 than in 2013. Finally, when looking at the measurement dates in Table B1, I find it surprising that most are taken within the last week of the month but for some unknown reason (which could be technical, but it is not communicated), the October 2013 data was measured on the 8th, which is not consistent with other data points. Given these comments, it is hard to justify treating the 2013 and 2023 months on an equal basis which is why I believe that the authors could improve the overall readability by sharing their reasoning behind choosing this particular dataset. It would be interesting to know if they are aiming at studying inter-annual or solely seasonal variability, in which case they would probably need to justify why they look at data taken 10 years apart. However, having such data could still be a strength if more was said about the evolution of some metrics over this decade.

The selection of the ERT dataset is explained more explicit in the revised manuscript. As this was also addressed by Referee 2, we include here our response (A12):

*Since the novelty of our manuscript is to reveal pressurised water flow during the thawing period, we refrained from a detail interpretation of the decadal permafrost change. Our initial motivation to include the data from 2013 was to cover all seasons, including the frost period, which was logistically not feasible in 2023, but would probably have yielded comparable results. We included a short paragraph in the discussion section 5.2 to clarify our choice of ERT data and suggest to refraining from a detailed interpretation of the decadal permafrost change, since we did not present ERT measurements from the same seasonal periods nor a decadal recording of borehole temperature or piezometric data (Figure 8). In addition, the ERT measurements from 2023 show that the signal is influenced by the water content of the rockwall, greatly affecting the detection of permafrost and, thus, making it impracticable to draw conclusions about its evolution.*

(Line 395-399): *"[…] Consequently, we refrained from detailed interpretation of the inverted tomograms from February-May 2013 and September-December* **2013 but included the ERT data to cover all season. We assume that a repeat of the ERT measurements in 2023 during the frost period would have yielded comparable results, as borehole temperature show frozen subsurface conditions from January-June and from October-December (Fig. 7), during which no thermal anomalies or irregularities were observed (Fig. 5).**"

The differences in the ERT results from September in 2013 and 2023 is likely driven by varying atmospheric conditions before the respective measurements. This was also already responded thoroughly in our response to Referee 2 (A12):

(Line 385-399): **"Atmospheric conditions vary slightly between years, affecting the timing of snow melt and hence the change in resistivity regime. The ERT results from June 2013 exhibit higher resistivity values in the upper part of the profile (> 64 kΩm) compared to the ERT measurement in 2023 (Fig. 7), probably due to colder rock and atmospheric conditions prior to the measurement (Fig. A1).** *The penetration depth of the current flow into the subsurface depends on the characteristics of the top layer and may vary seasonally. Dry and frozen conditions can impede current flow, while water-saturated conditions might trap current*

*flow, resulting in an attenuated current flow into deeper layers (Loke, 2022). Poor electrode coupling is often associated with frozen conditions and ice-filled fractures and cracks from autumn to spring, which cause noisy data and can lead to inversion artifacts represented by a high RMS error. **This phenomenon was observed, for example, in September 2013 vs 2023, where cold air temperatures likely caused freezing of the rock surface layer prior to the ERT measurement (Fig. A1), resulting in high contact resistance and an inability to resolve the long-lasting summer thermal signal in greater depths.***"

The decision to include the ERT measurement from October 8[th] was based on technical reasons. However, we believe that ERT observations at the end of October would have yield comparable results. This assumption based on the marginal differences observed in the ERT tomograms from September to December 2013 (Figure 7). Since we refrained from a detailed interpretation of the ERT measurements during winter due to the potentially reduced data quality (see Line 395-396), we did not further scrutinize the slight variation in observation dates during the frozen season, but we will point it out in the caption of the Table B1 in the revised manuscript.

Finally, the strength of combining the ERT measurements with borehole temperature data is precisely to be able to produce a plot like Figure 7b, providing some elements of proof of the presence of pressurized water flow. To me, this is the main message of the paper, and I believe it goes slightly unnoticed in the current layout. I would suggest emphasizing this result and providing more explanation of the processes at hand and the reasoning underpinning the conclusion.

The primary message of our manuscript is the detection of pressurised fluid flow during the thawing season. To enhance this point, we incorporated new piezometric measurements, which underpin our findings of the borehole temperature and ERT analysis. Additionally, we slightly modified the conclusion (see A15 in the response to Referee 2) to underscore our main message of the paper.

Overall comments on Figures and Tables:

- The axis labels are not centered, and not capitalized.
  Since the journal does not specify guidelines for axis label formatting, we have chosen our own layout.

- The Tables include some repetitions in the units, some confusing symbols, and some labels not previously introduced.
  We have carefully reviewed the Tables and removed the repetition of the unit in Table B1 (500**V**) and explained the parameter of the bilinear relation in Table A1 (**a+by**). However, we could not identify confusing symbols as we used only common declarations of parameters.

- Some text should accompany the Figures and Tables of the Appendix.
  The Figures and Tables in the Appendix are either explained in detail within the corresponding paragraphs of the main text or have their findings described directly in the Appendix. However, we provided a more detailed description for Table B1.

- Not all Figures are referenced, and the order of the Figures in the Appendix does not represent their reference order from the main text.
  We included the reference to Figure B3 in the main text and modified the order of the Figures and Tables in the Appendix.

Some extra comments:

- Line 100: Is there a particular reason behind this choice of diameter? Could you comment on how the relation could potentially change with a different diameter?
  The choice of diameter was based on technical constrains imposed by the drill bits used for extracting the core samples. However, we assume that using a different diameter of the core samples for the laboratory calibrations would not have yielded to significant different temperature-resistivity relationships.

- Line 143: What about the weather conditions in 2013? I believe it would be interesting to present some weather data in a table, say more about the air temperature, talk about precipitation, snow etc.
  See comment above.

- Line 163: The ERT doesn't give any information at depth below x = 0m, so could you please clarify why you decided to place B1 at the beginning of the survey line? A short sentence motivating the geometry of the survey would be interesting for the reader.
  The initial placement of boreholes was primarily intended for monitoring rock temperatures and was determined independently from the ERT monitoring. However, since both boreholes were drilled along the profile, we used them to describe the temperature regime of the investigated rock wall and, specifically, borehole B2 to directly link the resistivity values with rock temperature.

- Line 191 / Paragraph 4.2: In relation to previous comments, it might be interesting and even needed to add a paragraph studying the inter-annual variations.
  See comment above.

- Line 232: From Fig. 6, it seems to me that thermal anomalies are identified with thermal rate of change as low as $10^{-3}$ °C/10min. This corresponds to a difference of $1.2 \times 10^{-2}$ °C over an averaging window of 2h, which is an order of magnitude less than the claimed threshold of ~0.2°C over that same period above which heat transfer becomes non-conductive. Could you please provide more information here and clarify the agreement between the Figure and these statements?
  We modified the sentence to: *"[…] which exhibit a temperature rise of **up to 0.7 °C in less than 2 hours".***

- Figure 6 / B1 / 15m: It is mentioned that there are 'notable changes in the quasi-sinusoidal pattern since 2020' but I believe the reader would benefit from an explanation of the underlying cause for such a change.
  Please consider the detailed response to Referee 2 (A13).

- Line 273: It is surprising to read this sentence about the piezoemetric measurements without context. I would kindly suggest that the authors add some context and most importantly, present some data.
  See comment above and consider our detailed response to Referee 1 (A2) and 2 (A13) about the piezometric observations.

---

## Author Response (AR2)

Technical University
of Munich

[Figure]

TUM School of Engineering
and Design

Chair of Landslide Research

Maike Offer
Phone: +49 89 289 25867
**maike.offer@tum.de**

Technical University of Munich · Chair of Landslide Research
Arcisstr. 21 · 80333 Munich · Germany

The Cryosphere
**Dr. Teddi Herring**

Munich, November 05, 2024

**Author's response (egusphere-2024-893)**

Dear Dr. Teddi Herring,

Thank you for your positive feedback on our revised manuscript (egusphere-2024-893) entitled *"Pressurised water flow in fractured permafrost rocks revealed by borehole temperature, electrical resistivity tomography, and piezometer pressure",* which requires only minor revisions.

We have carefully answered all the comments from the two reviewers and slightly changed the manuscript. Please find attached our detailed point-by-point responses and our suggested changes.

We look forward to hearing from you.

Kind regards,

Maike Offer (on behalf of all authors)
* * *
**Reply to Referee #3**

Dear Referee #3,
Thank you for taking the time to review our revised manuscript entitled *"Pressurised water flow in fractured permafrost rocks revealed by borehole temperature, electrical resistivity tomography, and piezometric pressure" (egusphere-2024-893)* and acknowledging the difficult field work conditions in high-alpine environments. We greatly appreciate your positive feedback with only minor revisions. Our detailed point-by-point responses are given below, highlighted in blue, with proposed changes indicated in bold. Please note that the lines mentioned in our replies correspond to the second version of the revised manuscript.

Sincerely,
Maike Offer (on behalf of all authors)
* * *
The manuscript I have received was a revised version, accompanied by a detailed explanation of changes. This revision has been done carefully, and it is clear that the authors respond to the concerns of the reviewers. The study is after my opinion a timely and interesting case study demonstrating water flow in fractured bedrock affected by permafrost in a high-alpine setting. As far as I know, these data are novel and unique, and extremely difficult to obtain. I see the several shortcomings addressed by the reviewers related to different dates, instruments and inversion procedures, which would probably kill a study under very simple and controlled conditions. Here, we have a high-alpine environment, where such shortcomings have to be accepted.

So, in summary, I would recommend publication of the paper, after some minor revisions (see below). The revised manuscript reads well and is well structured. The figures are illustrative, and I think the study advances our knowledge on high-mountain permafrost processes in a time of global climate change. I have gathered some comments below, which you many address:

Abstract: The abstract is quite long and detailed and would benefit from making it a bit shorter.
(A1) We made the abstract a bit shorter and more readable.

Introduction: Consider deleting (or move somewhere else) the last sentences in the Introduction (from line 76), this is a sort of conclusion.
(A2) The last sentences were adapted and added in the first round of revisions in response to a comment from Referee#1 for a clearer articulation of the specific objectivities of our research (see our previous response, (A4)). To address the concerns of Referee#1, we propose keeping these sentences as they currently appear in the Introduction.

Chapter 3.2. This is a detailed and wordy description of the ERT, and the inversion process. Just give fundamentals and cite relevant literature. If you think this much important information, consider making a paragraph in the appendix.

(A3) To ensure that readers with varying backgrounds in ERT can benefit from our research, we initially included a detailed description of the ERT basics and inversion process. However, we agree that this level of detail might detract from the focus of the manuscript. Following your suggestion, we have moved this paragraph in the appendix.

Chapter 4.1. You may consider to provide some of data in the first paragraph of the chapter in a table. Much easier to understand.

(A4) We agree that the first paragraph of subchapter 4.1 contains many numerical details that might be challenging to follow. Rather than providing a new table, which would duplicate information already in Fig. 4, we have modified Fig. 4 to include the additional details from this paragraph (active layer thickness, depth of zero annual amplitude, thermal offset, and depth of 0 °C on the days of ERT measurements in 2023).

l. 258: This sentence should come later? ERT results not yet introduced here.

(A5) Thanks for your comment. We have retained the first part of the sentence in subchapter *4.1 Borehole temperature and thermal anomalies* ("All abrupt changes and irregularities in the borehole temperature occurred between late May and September.") but moved the context of the second part to subchapter *4.2 Seasonal variations in ERT* ("The most pronounced seasonal variation in electrical resistivity occurs in summer (Fig. B1), **coinciding with all abrupt changes and irregularities in borehole temperature (Fig. 7).**")

What is a "quasi-sinusoidal"

(A6) The term "quasi-sinusoidal" describes a pattern that closely resembles, but does not strictly conform to, a true sinusoidal curve. In the context of borehole temperature data, this term often refers to a repeating wave-like temperature pattern that approximates seasonal temperature cycles. As the term is already used in several studies, e.g. Hauck (2002), we decided not to further describe the term in the manuscript.

Hauck, C., Frozen ground monitoring using DC resistivity tomography, Geophys. Res. Lett., 29(21), 2016, doi:10.1029/2002GL014995, 2002.

Figure 5: make sure that the text in the figure is readable. Now for me just with a magnifying glass 😊

(A7) We have increased the size of the text in Fig. 5.

Chapter 4.3.: Is quite misplaced after the ERT results description. This calibration could be part of the Method-section under the ERT explanation, this is a procedure you do to get your results interpreted. Alternative, before the ERT result description.

(A8) In the first version of the submitted manuscript, the chapter of the laboratory calibration was presented as the first subchapter in both the methods and results. However, in the first round of revisions, Referee#2 suggested, "*I would have first presented the field before the laboratory calibration approach as the latter completes the former*" (see our previous response, (A6)). To address this concern, we restructured the subchapters to begin with

borehole temperature observations, followed by the field ERT measurements and concluding with laboratory calibrations. To maintain alignment with the comments of Referee#2, we suggest to retaining the current order of the subchapters.

Discussion: The chapter 5.2. about limitation should start the discussion. Now you discuss the results, then you say there are limitation, then you discuss further the results. So, move this to the start of the discussion. Check also for redundancies, a bit wordy the whole paragraph. The limitation is longer than the rest of the discussion, show clearly that the limitations do not hamper the value of the results.

(A9) Thank you for your suggestion. We believe that it is important to be transparent about limitations, but we think that starting with the study results will better capture the interest of the reader, especially since our focus is on the novel observation of pressurised water flow in permafrost rocks. We agree that the limitations section is detailed because of the range of measurement techniques we used – borehole temperature, electrical resistivity tomography, laboratory calibration, and piezometric pressure – all of which have uncertainties, especially in high-alpine environments. This section has been expanded in response to previous comments from reviewers and the editor to address uncertainties (e.g., different instruments and measurement years, atmospheric conditions, high contact resistance) in the measurements. To the best of our knowledge, we have highlighted all relevant limitations and explained why their impact on our results is minimal or why certain datasets (ERT 2013) have been excluded from the interpretation.

Conclusions: Consider deleting the last paragraph, does not give any new information.

(A10) We agree that the last paragraph of the conclusion does introduce new information; it was intended to outline future research steps in rockwall hydrology and to serve as a bridge to an upcoming manuscript. Therefore, we would like to keep this paragraph.

l. 319: change "paper" with "study"

(A11) →changed as suggested.

Fig. 8 caption: What do you mean with "this moving mean air temperature", who "moving", moving average? Over what period?

(A12) By "moving mean", we indeed refer to a "moving average", which calculates the mean over a sliding window of data points, moving incrementally over the dataset. We here used the definition of MATLAB.

In our case, as indicated in the caption of Fig. 8, we calculated the moving mean air temperature with a 2-hour sliding window. To further clarify, the air temperature data from the weather station was recorded at 10-minute intervals (as now included in subchapter 4.4: "The rapid rise in piezometric pressure correlates with days when the mean air temperature was above 0 °C, as measured at the weather station **in 10-minute intervals** on the nearby Gletscher Plateau (2.940 m asl)."

l. 334: why "inconcistent with Archie's law"?

(A13) The explanation why the massive decline in electrical resistivity from June to September is inconsistent with Archie's Law, unless pressurised water flow is assumed, is described from Line 325-330. We have slightly revised one sentence for clarity: "In thawed conditions, resistivity decreases for various rock types at a rate of ~ 2.9 ± 0.3 %/°C (Krautblatter, 2009), and according to our laboratory calibrations, by 4.5 ± 0.3 %/°C (Fig. 3, Table D1). Thus, **and considering Archie's Law,** a temperature warming from July to September (Fig. 9) in already fully saturated rock with constant porosity would not cause a significant further and rapid electrical resistivity decline. This can only occur if pressurised water flow contributes to additional hydraulic opening of fractures within days to weeks."

─────────────────────────────────────────────────────────

Dear Referee #2,

Thank you for reviewing of our revised manuscript *"Pressurised water flow in fractured permafrost rocks revealed by borehole temperature, electrical resistivity tomography, and piezometric pressure" (egusphere-2024-893)* and for the positive feedback indicating only minor revisions. Below, we provide detailed responses to your comments, highlighted in blue, with proposed changes in the revised manuscript indicated in bold. Please note that the lines mentioned in our replies correspond to the second version of the revised manuscript.

With kind regards,
Maike Offer (on behalf of all authors)
─────────────────────────────────────────────────────────

This second version of the paper now entitled "Pressurised water flow in fractured permafrost rocks revealed by borehole temperature, electrical resistivity tomography, and piezometric pressure" has adequately addressed the review's comment. Adding the piezometric data provides additional evidence and makes the paper more comprehensive. In its current form, it presents an interesting and unique combination of data showcasing a significant effort in integrating them despite their relative limitations. Additionally, the quality of the graphs is highly satisfactory. I now recommend this paper for publication after minor revisions. Minor comments:

- I would suggest adding the depth at which the "abrupt temperature changes are observed" in the abstract
(A1) Changed to: "[…] They further show abrupt temperature changes ($\sim$ 0.2-0.7 °C) **at 2, 3, and 5 m depth** during periods with enhanced water flow and temperature regime changes between 2016-2019 and 2020-2022 **at 10 and 15 m depth** […]"

- L42-43: this reference might also be interesting to complete the state of the art: Hugentobler, M., Loew, S., Aaron, J., Roques, C., and Oestreicher, N.: Borehole monitoring of thermo-hydro-mechanical rock slope processes adjacent to an actively retreating glacier, Geomorphology, 362, 107190, https://doi.org/10.1016/j.geomorph.2020.107190, 2020.
(A2) Hugentobler et al. (2020) provide valuable insights into thermo-hydro-mechanical processes on a rock slope affected by glacial retreat, monitored through deformation, groundwater pressure and temperature in three boreholes. However, the borehole measurements in this study indicate the absence of permafrost in the investigated slope. In the revised manuscript, especially in L39-44, we have focused on observations of piezometric pressures specifically in permafrost-affected regions. Therefore, we suggest not including the suggested reference in the introduction, but to clarify the knowledge gap: "[…] piezometric pressures have only been observed **in permafrost regions** on one rock glacier (Phillips et al., 2023; Bast et al., 2024) and in one open crack at shallow depth (Draebing et al., 2017). Direct observations of piezometric pressures in deep depths (> 10 m) have not yet been measured in permafrost-affected rockwalls, but remain crucial for understanding hydrological processes and thus for the prospective prediction of rock slope failures."

- L94: did you mean "installed in 2012"?
(A3) Thanks for identifying this mistake. We have corrected it to: **"[…] installed in 2012**".

- L241: this sentence is a little bit confusing: did you interpolate borehole temperature from ERT measurements?? Or did you mean temperature measurements during ERT measurements.
(A4) We here refer to borehole temperature on the days when ERT measurements were taken. To clarify, we have revised the sentence to: "**Interpolated borehole temperature recorded on the days of the ERT measurements in 2023** […]".

- L260: figure 7 is cited before figure 6
(A5) As we moved the corresponding sentences to subchapter 4.2 (see our response (A4) to Referee #3), Fig. 6 is now cited before Fig. 7.

- L269: I would rather say "pluriannual changes" rather than "long term" as it is < 5-year trend
(A6) Changed as suggested.

- L275, Fig. 6: the paper should be checked by a native speaker. Writing "in" 10 m depth sounds not appropriate to me. Wouldn't it be "at"?
(A7) We modified the caption of Fig. 6 and L264 to: "[…] **at** 10 m depth", and "[…] in July at 10 m depth […] and in September **at** 15 m depth […]".

Finally, I would recommend thoroughly reviewing the references, as some are not used appropriately: e.g. L247, Gruber et al., 2004 is not the paper that defines the DZAA; L340, Noetzli et al., 2007 doesn't address non-conductive heat transfers; L359: Magnin and Josnin, 2021, doesn't use ERT measurements. Those are only a few examples, but all references should be thoroughly reviewed.
(A8) We acknowledge that Gruber et al. (2004) refers to the depth of zero annual amplitude but does not define it. Since this term is commonly used within the permafrost community, we have removed the citation.
Noetzli et al. (2007) was cited to support that rapid changes in rock temperature cannot be explained solely by diffusive heat exchange (L330-332). This study demonstrates that temperature signals based solely on heat conduction take considerable time to penetrate deeper into a ridge with a north-south orientation, like the Kitzsteinhorn. Therefore, the rapid rock temperature changes cannot be explained by the models presented in Noetzli et al. (2007).
Magnin and Josnin (2021) was incorrectly cited twice in the same sentence. We corrected this to: "[…] as already shown by geophysical measurements (Krautblatter and Hauck, 2007; Krautblatter et al., 2010; Keuschnig et al., 2017) and by numerical approaches (Magnin and Josnin, 2021) […]".
We carefully reviewed all other references but could not find further mistakes.